# TwinMarket: A Scalable Behavioral and Social Simulation for Financial Markets

**Yuzhe Yang**[1,*] **Yifei Zhang**[1,2,*] **Minghao Wu**[1,2,*]
**Kaidi Zhang**[1] **Yunmiao Zhang**[2] **Honghai Yu**[2,3,†] **Yan Hu**[1,†] **Benyou Wang**[1,†]
[1] The Chinese University of Hong Kong, Shenzhen, [2] Nanjing University
[3] Jiangsu Key Laboratory of Digital Finance
{yuzheyang,minghaowu}@link.cuhk.edu.cn, {huyan,wangbenyou}@cuhk.edu.cn
yf_zhang@smail.nju.edu.cn, hhyu@nju.edu.cn
https://freedomintelligence.github.io/TwinMarket

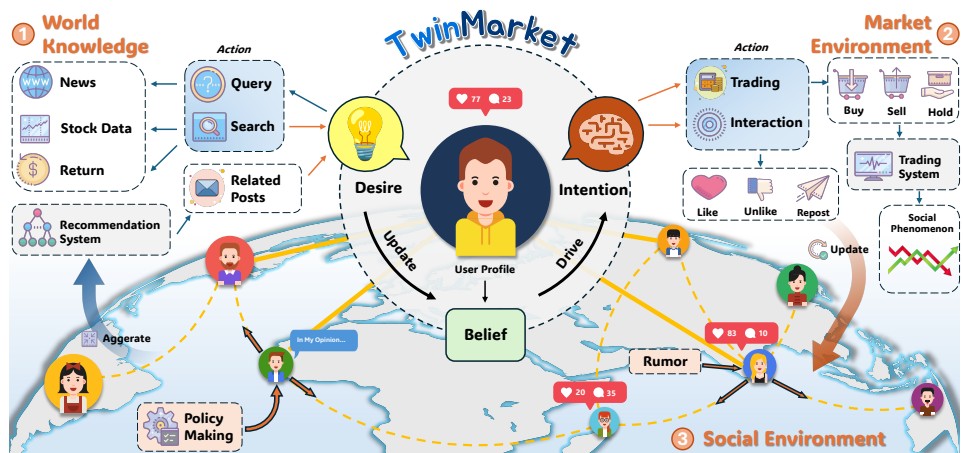

Figure 1: Overview of **TwinMarket** environment where each user has a unique persona within the social network, interacts with the environment in real-time, and influences it through their actions. This framework enables the study of emergent social phenomena.

## Abstract

The study of social emergence has long been a central focus in social science. Traditional modeling approaches, such as rule-based Agent-Based Models (ABMs), struggle to capture the diversity and complexity of human behavior, particularly the irrational factors emphasized in behavioral economics. Recently, large language model (LLM) agents have gained traction as simulation tools for modeling human behavior in social science and role-playing applications. Studies suggest that LLMs can account for cognitive biases, emotional fluctuations, and other non-rational influences, enabling more realistic simulations of socio-economic dynamics. In this work, we introduce **TwinMarket**, a novel multi-agent framework that leverages LLMs to simulate socio-economic systems. Specifically, we examine how individual behaviors, through interactions and feedback mechanisms, give rise to collective dynamics and emergent phenomena. Through experiments in a simulated stock market environment, we demonstrate how individual actions can trigger group behaviors, leading to emergent outcomes such as financial bubbles and recessions. Our approach provides valuable insights into the complex interplay between individual decision-making and collective socio-economic patterns.

---

*Equal contribution. †Corresponding authors.

39th Conference on Neural Information Processing Systems (NeurIPS 2025).

# 1 Introduction

Recent advances in large language models (LLMs) have showcased their remarkable capacity to capture complex patterns in language, decision-making, and social interactions [1, 2]. These capabilities have positioned LLMs as powerful instruments for agent-based simulations in the study of human behavior, supporting a wide range of social science applications, including conversational dynamics [3–6], emotional responses [7], and creative reasoning [8–10]. These developments open new frontiers for social science research, enabling the modeling of fine-grained individual behavior at the **micro-level** and the simulation of complex, emergent phenomena at the **macro-level**.

Financial markets, with their unique combination of high-resolution transactional data, observable collective phenomena *(e.g., herding behavior & market crashes)*, and direct socio-economic relevance, provide an ideal testbed for studying how micro-level behaviors propagate to macro-level dynamics [11, 12]. However, existing agent-based models (ABMs) in social simulation primarily rely on rule-based approaches, which often oversimplify human decision-making by assuming homogeneous agents and static behavioral rules [13, 14]. These oversimplifications fail to capture critical intricacies of real-world economic behavior and struggle to scale to market complexity, obscuring causal links from individual decisions to systemic emergence.

The study of social emergence phenomena, where **micro-level** individual behaviors aggregate to form **macro-level** collective outcomes, has long been a central focus in social science [15, 16]. However, current methods often struggle to capture the dynamic interplay between individual biases *(e.g., overconfidence)* and emergent market phenomena *(e.g., volatility clustering)* [17, 18]. Recent advances in LLMs offer a promising avenue to address this challenge, enabling the simulation of adaptive agents whose decisions are shaped by both private information and social interactions. Motivated by this potential, we propose TwinMarket, a novel multi-agent framework that leverages LLMs to simulate investor behavior in a stock market environment. By modeling investors that independently make investment decisions while interacting through a simulated social media platform, TwinMarket provides a unified perspective on **how micro-level behaviors drive macro-level market dynamics**.

The key innovation of TwinMarket is its use of the **Belief-Desire-Intention** (BDI) framework [19] to structure and visualize the cognitive processes of agents, providing a transparent and interpretable modeling approach. Additionally, we develop a social environment that enables agents to engage in dynamic information exchange and social influence processes, enabling the study of phenomena such as the emergence of opinion leaders and the dynamics of information cascades. Finally, by scaling our simulations to multiple financial assets and larger populations, we investigate how group size and interaction complexity affect collective behavior and emergent phenomena.

Our research makes three key contributions:

❶ **Real-World Alignment:** Our framework is grounded in established behavioral theories and calibrated with real-world data, ensuring theoretical rigor and realistic human behavior modeling. This enables us to accurately reproduce certain social phenomena, allowing us to further study the mechanisms behind these phenomena.

❷ **Dynamic Interaction Modeling:** By leveraging an LLM-based framework, we capture diverse human behaviors and their interactions, particularly in the context of information propagation. This allows us to model how opinions spread, influence decision-making, and dynamically shape collective market behaviors in real time.

❸ **Scalable Market Simulations:** We conduct large-scale simulations to analyze the impact of group size on market behavior, providing new insights into price fluctuations and emergent dynamics.

# 2 Background

Traditional ABMs face two major challenges in simulating complex social systems [20]: (1) customizing sophisticated economic decision-making for individual agents and (2) enabling agents to actively perceive and respond to external environmental changes. Recent research has made progress by incorporating LLMs into agent-based simulations. CompeteAI [21] introduces a framework for modeling competitive dynamics; however, its agents rely on predefined prompts, limiting their ability to simulate nuanced economic behavior. EconAgent [22] designs economic agents with perception,

reflection, and decision-making capabilities to model macroeconomic dynamics but overlooks the influence of multi-agent interactions on emergent macroeconomic phenomena. Similarly, ASFM [23], which models stock market responses to regulatory policies, fails to adequately account for the role of agent interactions and collective decision-making in shaping market trends.

In our TwinMarket framework that leverages LLM-based agents to simulate complex human behaviors, we include rational decision-making, technical analysis, and behavioral biases such as herding and overconfidence [24, 25]. By grounding our model in established behavioral theories [26–

Table 1: Existing financial agent systems.

| System | Agent Design | Scale / Interaction |
|--------|--------------|---------------------|
| **CompeteAI** | Fixed prompts | 2 agents, competitive |
| **EconAgent** | Demographic-based | 100 agents, no interaction |
| **ASFM** | Two strategies | N/A, no interaction |
| **TwinMarket** | Fully configurable | 1000+, rich interaction |

28] and calibrating it with real-world data, we ensure both theoretical rigor and empirical relevance. In TwinMarket, agents make investment decisions within a socially embedded environment where interactions occur through price dynamics and social media, shaping collective market behavior via information propagation, behavioral imitation, and interaction mechanisms.

This work demonstrates the potential of LLMs to bridge computational simulations and behavioral science by systematically modeling **micro-level** cognitive processes and their **macro-level** societal effects. By integrating LLMs with established theoretical frameworks, TwinMarket provides an experimental platform to study how individual decision-making aggregates into emergent social and economic phenomena. Our findings illustrate **how LLMs can simulate real-world behaviors, validate behavioral theories, and uncover mechanisms underlying social emergence**, advancing the academic understanding of complex human systems.

# 3 TwinMarket

## 3.1 Overall Design of TwinMarket

The design philosophy of TwinMarket emphasizes both **micro-level** and **macro-level** simulation. At the **micro-level**, we focus on modeling user behavior, as illustrated in Figure 2(a). This entire process is driven by BDI framework, which we will discuss in Section 3.2. At the **macro-level**, as illustrated in Figure 2(b), we simulate two core infrastructures of real-world financial ecosystem: an order-driven trading system that settles transactions and updates the stock index accordingly (detailed in Appendix E.5), and a social media environment that personalizes information delivery based on user behavior while enabling social interaction among agents.

Additionally, TwinMarket is a data-driven platform that leverages real-world data sources to construct a realistic and credible simulation environment, as summarized in Table 2. Our synthetic user data is sampled based on behavioral patterns observed from real users, and enriched with actual stock data, news, and company announcements. This setup enables simulated users to make informed decisions, allowing us to observe the emergence of macro-level market phenomena from individual interactions.

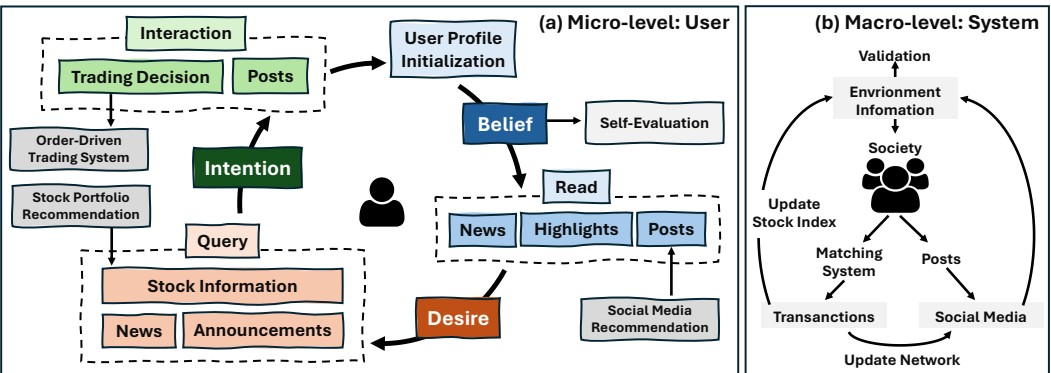

Figure 2: The simulation workflow driven by the **Belief** - **Desire** - **Intention** framework. The micro-level simulation focuses on modeling individual user behavior, while the macro-level simulation addresses the dynamics of the social media platform and trading system.

Table 2: Summary of data sources for TwinMarket. Further details about our data sources are provided in Appendix C.

| Usage | Source | Quantity or Time Period | Data Type |
|---|---|---|---|
| **User Profile Initialization** | Xueqiu[1] | 639 users
11,965 transactions | User Profile
Transaction Details |
| **Stock Recommendation** | Guba Stock Forum[2] | 83,246 transactions | Transaction Details |
| **Update Fundamental Stock Data** | CSMAR[3] | Jan 2023 - Dec 2023 | Stock Data[4] |
| **World Knowledge Database** | Sina[5] & 10jqka[6]
CNINFO[7] | Jan 2023 - Dec 2023 | News Articles
Company Announcements |

[1] https://xueqiu.com/    [2] https://guba.eastmoney.com/    [3] https://data.csmar.com/
[4] We choose the 50 largest and most liquid stocks on the Shanghai Stock Exchange (known as the SSE 50 index).
[5] https://www.sina.com.cn/    [6] https://www.10jqka.com.cn/    [7] http://www.cninfo.com.cn/new/index.jsp

## 3.2 Micro-Level Simulation: BDI-driven User Modeling

In user modeling, the BDI framework plays a central role in simulating the cognitive processes of users. It enables each agent to perceive the market environment, integrate that information with its individual profile, and form a unique chain of reasoning that guides decision-making. Figure 2(a) shows how an LLM-based agent operates in a daily loop: receiving news and posts, querying from knowledge base, making trading decisions, and interacting via social platforms. Each action in this loop is governed by the agent's beliefs, desires, and intentions, enabling nuanced and individualized behavioral responses. A formal definition of the BDI components and the agent's daily operational cycle is provided in Appendix E.2.

**Belief.** Belief provides the dynamic foundation for how an agent perceives and interprets its environment. To ensure agents follow consistent personas, we inject system prompts that encode individualized traits. Using real transaction data and user profiles from Table 2, we first estimate behavioral biases and then extract key characteristics such as demographics and investment styles. These features are used to synthesize user profiles, forming the basis of our agent society. The distribution of these synthetic profiles closely aligns with findings[2] from prior financial research [29]. As user profiles are initialized, beliefs encoded as system prompts guide agents to read and interpret news and social media posts. These perceived signals, grounded in agent traits, shape their evolving understanding of the environment and serve as the cognitive basis for subsequent decision-making.

After completing all actions for the day, each agent evaluates its behavior based on both its actions and the environmental feedback it received. Through this process, the agent dynamically updates its beliefs, reflecting an evolving understanding and interpretation of the market. This captures the notion that a user's perception and judgment of the environment are not static but shaped continuously by experience and new information. To track belief dynamics and explore micro-to-macro linkages, agents output a sentiment score[3] at each update step.

**Desires.** Desires represent the goals or objectives an agent aims to achieve based on its **beliefs**. While **beliefs** provide an understanding of the environment, agents autonomously generate queries to retrieve relevant stock and news information, enhancing their comprehension of the market context.

**Intentions.** Intentions represent the committed plans or actions that an agent chooses to pursue among its possible desires. While **desires** reflect what an agent wants, intentions define what it actively decides to do, given its current **beliefs** and constraints. These **intentions** drive the agent to make final trading decisions and to express or amplify opinions through posting or reposting content on social media.

## 3.3 Macro-Level Simulation: Dynamic Social Network

The relationship between micro-level agent behavior and macro-level market dynamics is central to understanding complex financial systems [30]. The market we design is inherently dynamic, driven

---

[2] Order-level transaction data is sensitive, and our goal is to show that the data we collect is reliable, especially compared to prior studies with access to proprietary firm data.

[3] The sentiment score ranges from 1 to 5 and is computed as the average of five self-assessed dimensions, based on user's views on: economic fundamentals, market valuation levels, short-term market trends, the sentiment of surrounding investors, and self-confidence, as detailed in Appendix E.1.

by agents whose behaviors evolve over time. This requires the system to be both adaptive and scalable, capable of responding to shifts in individual behavior and interaction patterns. To achieve this, we follow two key principles. First, the graph structure must evolve dynamically to reflect changes in agent connectivity. Second, agents must be able to aggregate and interpret environmental information through their networked relationships.

**Constructing Dynamic Social Network.**
We model the social network as a dynamic graph $\mathcal{G} = (\mathcal{V}, \mathcal{E})$, where $\mathcal{V}$ is the set of users and $\mathcal{E}$ denotes weighted edges that capture similarity in trading behavior. Each user trades industry-level stock index $I = \{i_1, \ldots, i_m\}$, and we assume that users with more similar trading patterns are more likely to interact [31, 32]. To reflect temporal dynamics, we apply exponential time decay to each trading record $T(u, i, t)$, resulting in a time-weighted trading intensity $w(u, i) = \sum_{t \in T(u,i)} e^{-\lambda \cdot \Delta t}$, where $\Delta t$ is the time elapsed since trade time $t$. We then compute the weighted Jaccard similarity[4] between users to determine the strength of social connections [33]. We provide a sensitivity analysis of graph construction parameters in Appendix B.1. Figure 3 shows the dynamic social network as a chord diagram. Colors indicate user groups with similar industry preferences *(e.g., Manufacturing, Finance Service)*, and chord thickness reflects the strength of trading similarity between groups.

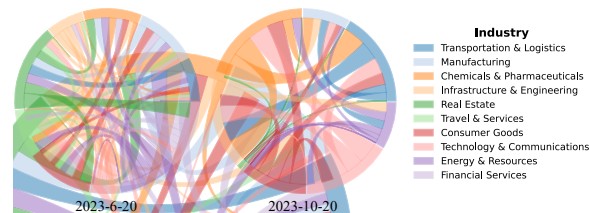

Figure 3: Chord diagram of user dynamic social network.

**Information Aggregation Mechanism.**
To control each agent's perception field and ensure exposure to relevant information, we design an aggregation mechanism based on the dynamic social network. As illustrated in Figure 4, information forms propagation chains as it spreads through the network, and the information each agent receives is restricted by its perception field. For a target user $u_t \in \mathcal{V}$, we consider neighbors $u \in \mathcal{N}(u_t)$ with similarity $S(u_t, u) > \tau$. For each such user, we retrieve their posts $P(u)$ within a time window $[t_s, t_e]$, where each post $p \in P(u)$ contains metadata such as timestamp $t_p$, upvotes $u_p$, and downvotes $d_p$. We compute a hot score $h(p)$ for each post to balance popularity and recency:

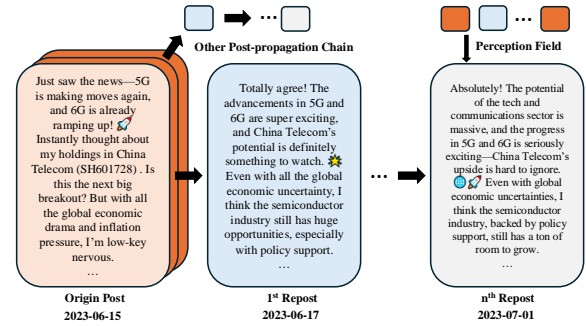

Figure 4: Information propagation in network.

$$h(p) = \frac{\log_{10}(u_p - d_p + 1)}{(T_p + 1)^{1.8}}, \quad T_p = \max\left(\frac{t_c - t_p}{\alpha}, \epsilon\right),$$

where $t_c$ is the current time, $\epsilon$ prevents division by zero, and $\alpha$ is a time normalization constant controlling decay rate. As illustrated in Figure 5, we set $\alpha$ such that a post's influence typically lasts around two weeks before its score significantly diminishes. Posts from all qualified neighbors are ranked by their hot scores, and each user is recommended the top $k$ posts per day. This mechanism enables personalized and dynamic information exposure shaped by social similarity and engagement dynamics. More technical details about information on social media are further discussed in Appendix E.3.

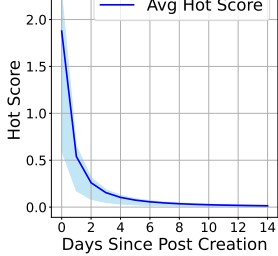

Figure 5: Hot score decay: Post influence lasts 2 weeks.

## 4 Experimental Validation

To evaluate the realism and reliability of our simulation framework, we conduct a series of experiments comparing the outcomes generated by TwinMarket with real-world financial and behavioral data. Our

---

[4]We use a generalized Jaccard similarity for real-valued vectors, calculated by: $S = \sum_i \min(w_1, w_2) / \sum_i \max(w_1, w_2)$.

validations are performed both in micro and macro level, which are conducted on a simulated social network with 100 user agents, where each agent is powered by GPT-4o. The simulation runs from 2023-6-15, and lasts for five months.

## 4.1 Micro-level Validation

We validate the micro-level realism of TwinMarket by examining emergent behavioral and economic patterns, with qualitative examples discussed in Appendix B.5.

**Inequality in the Stock Market.** TwinMarket exhibits emergent patterns of wealth inequality, demonstrating its micro-level fidelity. As shown in Figure 6, the Gini coefficient displays a clear upward trend, accompanied by a growing wealth share for the top 10% of agents and a declining share for the bottom 50%. This self-organizing wealth divergence arises from the heterogeneity and interactions among agents, echoing established findings in the economics literature [34]. It highlights the model's capacity to generate realistic socio-economic dynamics from its micro-level specifications.

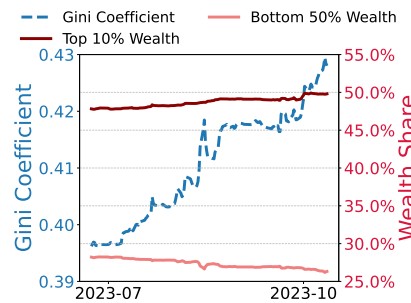

Figure 6: Rising Gini coefficient, indicating widening wealth inequality.

**Trading Activity and Returns.** Table 3 shows the top 10% of agents (ranked by return) achieved an average return of 6.65% with a modest turnover rate of 4.02%. In contrast, the bottom 50% showed a higher average turnover rate of 7.03% but suffered a substantial negative return of -10.52%. This pattern reflects a well-documented phenomenon among retail investors, where higher trading

Table 3: Agent performance vs. Trading frequency in TwinMarket

| Performance | Avg. Turnover | Avg. Return |
|---|---|---|
| Top 10% | 4.02% | 6.65% |
| Bottom 50% | 7.03% | -10.52% |

frequency even before accounting for transaction costs correlates with poorer performance [35]. Such behavioral consistency further reinforces the micro-level validity of TwinMarket.

## 4.2 Macro-level Validation

To assess TwinMarket's ability to replicate realistic market dynamics, we compare its simulated outputs against historical data by examining four widely recognized stylized facts. These are empirical statistical regularities consistently observed in real-world financial markets, and are commonly used as benchmarks for validating the realism of market simulation models [36].

**Stylized Fact I. Fat-tailed distributions [37]:** *Big market moves happen more often than what a normal distribution would suggest. This means extreme ups and downs are not as rare as we might expect.*

Figure 7 shows TwinMarket's price change distribution (**Log Return** on its x-axis) mirroring real-world markets by exhibiting **fat tails**. While most daily price changes are small (forming its sharp central peak), large, unexpected surges or plunges (the **heavy tails**) occur more frequently than a normal distribution would predict, capturing actual market risk. This realism stems from our autonomous agents, which, based on BDI cognitive frameworks, process information heterogeneously, leading to diverse and potentially conflicting sentiments. These divergent sentiments then propagate through the integrated social network, becoming amplified and polarized, which in turn directly leads to extreme market outcomes.

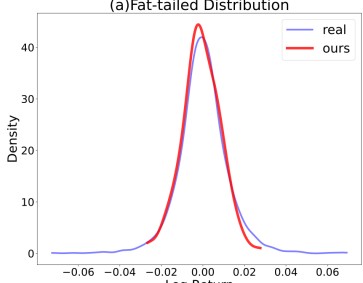

Figure 7: Fat-tailed price changes: more extreme movements.

**Stylized Fact II. Leverage Effect [38]:** *When prices go down, the market often becomes more volatile afterward. This means bad news makes the market more nervous than good news.*

Figure 8 shows TwinMarket capturing the leverage effect, where volatility responds asymmetrically. Specifically, initial price drops (x-axis **Negative Returns**) provoke larger subsequent drops (y-axis **Lagged Negative Returns**), thereby amplifying market volatility. This realistic asymmetry is a direct consequence of our LLM agents' design; their BDI cognitive architectures are intentionally embedded with behavioral biases, such as loss aversion. This design makes them, akin to humans, inherently more sensitive to adverse signals. The integrated social media platform further amplifies the spread of negative sentiment, resulting in more widespread, synchronized, and potentially destabilizing collective trading behavior.

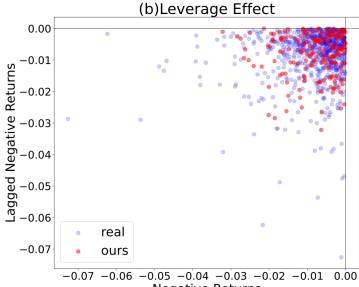

Figure 8: Initial price drops can trigger subsequent price drops.

**Stylized Fact III. Volume-Return Relationship [39]:** *Larger price changes usually happen when more people are trading, showing a positive link between trading volume and return size.*

Figure 9 shows TwinMarket's ability to replicate the volume-return relationship. Specifically, it shows that substantial increases in trading activity (x-axis **Volume Change**) are typically accompanied by significant market price fluctuations (y-axis **Log Return**). This co-movement is a key indicator, as in real markets, of collective behaviors such as herding among participants. In TwinMarket, the emergence of this co-movement provides compelling evidence demonstrates that our LLM agents, through their interactions, successfully generate collective behavioral patterns consistent with those observed in actual financial market dynamics.

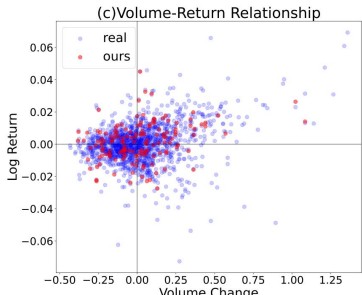

Figure 9: Corr. between trading volume & price changes.

**Stylized Fact IV. Volatility Clustering [40]:** *When the market is volatile, it often stays volatile for a while. When it's calm, it usually stays calm too. This means that big or small price changes tend to come together, not appear randomly.*

Figure 10 shows TwinMarket successfully replicating volatility clustering: periods of large price swings (high **Volatility** on y-axis) tend to cluster together, as do periods of small swings (low **Volatility**). Both simulated and real volatility series clearly show these alternating groupings of high and low market turbulence over time. This emergent persistence arises as impactful market-moving events, processed via the BDI cognitive architecture, induce sustained heightened belief sensitivity. The social network then perpetuates these collective cognitive states by circulating related discussions or sentiments, prolonging periods of either erratic or stable trading.

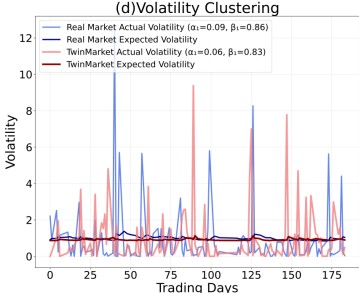

Figure 10: Big price swings group together; small ones too.

**ABM Baselines Comparison.** We compare Twin-Market against two transitional agent-based models: the HPM Model [41], which captures investor strategy adaptation through herding behavior, predispositions, and reactions to price misalignments; and the BH Model [42, 20], which incorporates hyperbolic discounting and stochastic volatility to reflect heterogeneous investor behavior. As summarized in Table 4, TwinMarket consistently generates outputs that align

Table 4: Comparing Stylized Facts (SF) across Real Data, TwinMarket, and ABM baselines.

| System | SF I | SF II | SF III | SF IV |
|---|---|---|---|---|
| Real Data | 7.26 | 0.14 | $p<0.01$ | 0.95 |
| **TwinMarket** | **5.24** | **0.11** | $p<0.01$ | **0.89** |
| ABM-HPM | 4.47 | 0.05 | $p<0.01$ | 0.82 |
| ABM-BH | 4.99 | 0.19 | $p<0.01$ | 0.72 |

more closely with real-world market statistics across all four stylized facts. This demonstrates its superior ability to replicate fundamental financial dynamics, thereby validating the empirical robustness of our simulation framework. These results were further validated for reproducibility and stability, as detailed in Appendix B.3.

The four stylized facts (SF) in Table 4 are quantified as follows: **(1) Kurtosis** (SF I): Measures tail heaviness; values greater than 3 indicate fat-tailed return distributions. **(2) Negative Return Autocorrelation** (SF II): Captures asymmetric responses to negative shocks. **(3) Volume Return Correlation** (SF III): A significant $p$-value ($< 0.01$) suggests stronger volume–volatility linkage. **(4) GARCH Volatility Clustering** (SF IV): The sum $\alpha + \beta \approx 1$ reflects persistent volatility, a hallmark of financial time series [43].

## 5 Emergence Analysis

Building upon the empirical validation of our simulation results, we have demonstrated that Twin-Market can generate behavior and market patterns consistent with real-world financial and social dynamics. To further investigate how micro-level agent behaviors give rise to macro-level phenomena, we conduct an extended set of experiments aimed at uncovering the mechanisms of social and market emergence. These analyses focus on the interplay between individual belief updates, local interactions, and global outcomes, offering insight into how localized decision rules and information asymmetries can lead to systemic patterns such as bubbles, polarization, and volatility.

### 5.1 Micro-level Emergence: Self-fulfilling Prophecy

A self-fulfilling prophecy describes a process where individual expectations collectively influence behaviors in ways that ultimately validate the original expectations [44]. This mechanism exemplifies micro-level emergence: localized belief updates and actions aggregate into systemic market outcomes. In our BDI framework, agents continuously revise beliefs, which guide their trading decisions and shape market dynamics. Twin-Market simulations reveal that early optimism amplifies buying pressure, driving prices upward and reinforcing collective confidence. This positive feedback loop detaches prices from fundamentals until a sharp correction ensues. As shown in Figure 11, belief and price co-evolve, replicating the speculative boom-bust cycle [45] and highlighting how collective belief formation drives macro-level patterns.

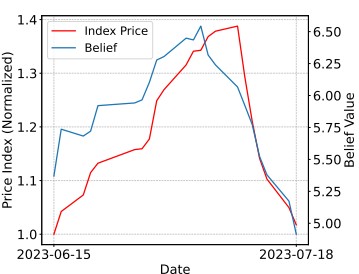

Figure 11: Simultaneous rise and fall of the price index and belief, illustrating their strong correlation.

### 5.2 Macro-level Emergence: Information Propagation

Previous work highlights how information propagation produces market-wide echo effects [46, 47]. To probe the same mechanism, we identify the day's most salient news items and deliver them directly to users with the highest network centrality, reproducing the information asymmetry characteristic of real markets. We then rerun the experiment with a parallel set of exaggerated, rumor-like headlines (We list some examples in Appendix E.4.) to stress-test the system. Observing how these targeted, micro-level interventions cascade through the network and reshape aggregate prices and volumes allows us to treat information diffusion as a source of macro-level emergence: the collective dynamics that arise cannot be inferred by examining any single agent in isolation, but only by tracking the system-wide response to the injected signals.

**Intimation & Polarization.** In TwinMarket, graph edges represent behavioral similarity, meaning users closely connected exhibit stronger intimation effects. As shown in Figure 12, high-degree users receive more upvotes, indicating their greater influence within the network. These **opinion leaders** [48] shape decision-making, fostering homogeneity and leading to polarization, as depicted in Figure 13 (further explored in Appendix B.4). The network's adjacency matrix highlights clusters

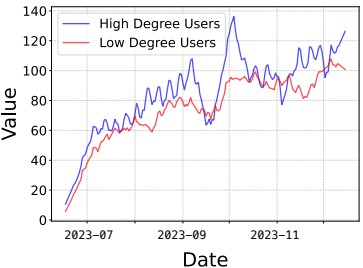

Figure 12: Average likes trends among different centrality users.

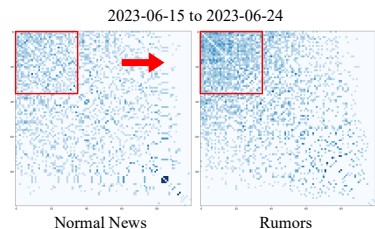

Figure 13: Heatmap of the user trading similarity adjacency matrix. Higher density indicates greater behavioral similarity.

of trading similarity, further intensified under rumor conditions. This suggests that biased or extreme information spreads more rapidly in tightly connected groups, reinforcing polarization [49, 50].

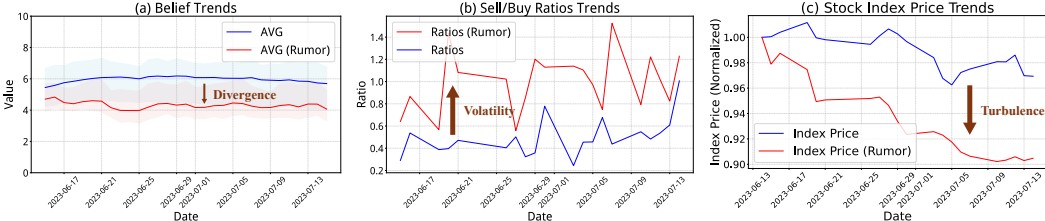

Figure 14: Rumor propagation impacts user beliefs and market trends. (a) Users' belief scores decline under rumor. (b) Users exhibit a stronger tendency to sell, and Sell/Buy ratio increase. (c) The collective reaction to rumors triggers a market downturn, causing stock prices drop-down.

**Belief Divergence.** Figure 14(a) illustrates how belief trajectories diverge over time between the control and rumor-exposed groups.[5] Replacing neutral news with negative rumors pushes expectations downward and reinforces echo chamber dynamics [51, 52]. Rumor-affected users become increasingly pessimistic, interact primarily with like-minded peers, and distance themselves from the broader population [53, 54]. This process intensifies belief heterogeneity and entrenches informational silos in the market.

**Trading Volatility.** Rumors do not merely shift beliefs, they also alter market behavior. As valuations decline, agents transition from buying to selling, triggering sharp changes in order flow [55, 56]. Figure 14(b) shows that the sell-to-buy ratio nearly doubles under rumor exposure, rising from 0.495 to 0.997. This dramatic increase reflects panic-driven responses to uncertainty and exemplifies how misinformation can amplify volatility in financial systems.

**Market Turbulence.** The cumulative effect of belief erosion and reactive trading destabilizes the broader market. Figure 14(c) shows that while prices remain stable in the control setting, rumor-exposed markets experience a pronounced decline. This reflects overreaction and herding behavior under uncertainty [57]. As negative sentiment spreads, confidence collapses, triggering self-reinforcing downturns that destabilize asset values and contribute to systemic fragility.

## 6 Ablation Study

To assess the contributions of TwinMarket's core design elements, we conduct an ablation study comparing the full model against two variants: (1) **w/o BDI**, which removes the belief-desire-intention module, making agents purely reactive; and (2) **w/o Hetero.**, which eliminates agent heterogeneity by assigning uniform strategies and biases. As shown in Table 5, removing either component degrades performance across predictive accuracy (RMSE, MAE, correlation) and financial realism (kurtosis, leverage effect, GARCH parameters).

Table 5: Ablation study on the effects of the BDI framework and agent heterogeneity. ↑: higher is better, ↓: lower is better. Corr., RMSE, and MAE are computed on the daily normalized index price series between simulated outcomes and real-world data.

| Model Variant | RMSE ↓ | MAE ↓ | Corr. ↑ | Kurtosis ↑ | Lev. Effect ↑ | GARCH $\alpha$ ↑ | GARCH $\beta$ ↑ |
|---|---|---|---|---|---|---|---|
| **TwinMarket** | **0.02** | **0.02** | **0.77** | **5.24** | **0.11** | **0.06** | **0.83** |
| TwinMarket w/o BDI | 0.07 | 0.05 | 0.34 | 4.25 | 0.05 | 0.03 | 0.76 |
| TwinMarket w/o Hetero. | 0.09 | 0.08 | -0.61 | 3.58 | 0.13 | 0.05 | 0.85 |

The BDI ablation notably weakens the model's ability to generate stylized facts, such as fat tails and asymmetric volatility, highlighting the importance of cognitive reasoning in agent behavior. In contrast, removing heterogeneity leads to a sharper collapse in realism, with near-Gaussian returns and negative correlation, underscoring the necessity of strategic diversity for emergent market complexity. Together, these results confirm that both BDI framework and agent heterogeneity are essential for replicating real-world market dynamics. Additional ablation studies appear in Appendix B.2.

---

[5]On average, rumor-exposed agents assigned asset valuations 27.5% lower than those in the baseline.

# 7 Scalability

To assess the potential of data-driven scaling as a performance enhancement strategy, we investigate how varying levels of agent activation affect TwinMarket's predictive performance. We simulate markets with 10%, 20%, 40%, and 80% of agents actively trading each day,[6] while keeping other conditions fixed.

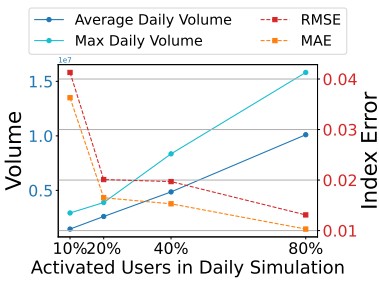

Figure 15:
Scaling law: Performance ∝ Volume.

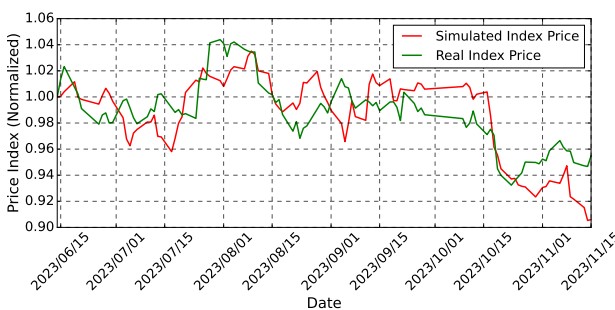

Figure 16: Simulation result of 1,000 agents.

As illustrated in Figure 15, increased agent activation yields not only enhanced trading volume but, more critically, a monotonic reduction in predictive error metrics (RMSE and MAE), revealing a robust scaling relationship between market participation and model fidelity. This empirical finding aligns with theoretical predictions from collective intelligence literature that expanded agent populations engender richer information aggregation environments wherein price discovery mechanisms become increasingly robust.

Figure 16 provides further empirical validation of this scaling principle, demonstrating that TwinMarket maintains both behavioral realism and computational tractability when scaled to 1,000 agents over extended temporal horizons. Collectively, these results show that sufficiently large-scale micro-level agent interactions can generate macro-level market phenomena that exhibit the statistical regularities and emergent properties characteristic of empirical financial systems.

# 8 Conclusion and Future Work

In this paper, we investigate social emergence phenomena through LLM-driven agent methods. We introduce TwinMarket, a novel multi-agent framework that leverages LLMs to simulate investor behavior in a stock market environment. Our findings demonstrate that LLMs can effectively model real-world behaviors, validate behavioral theories, and reveal underlying mechanisms of social emergence, contributing to a deeper understanding of complex human systems.

The implications of this work also extend to the machine learning community. Methodologically, we demonstrate that BDI-structured LLM agents can achieve higher behavioral fidelity than traditional models, offering a blueprint for more realistic multi-agent systems. Furthermore, the framework serves as a powerful engine for generating high-fidelity synthetic data, valuable for training and evaluating downstream models, such as reinforcement learning agents.

Building on these contributions, our future work will aim to enhance the framework's scope and realism. We will address the limitations of the current study, which focuses on the unique context of the China A-share market and employs simplified mechanics such as a single daily call auction and a zero-sum environment. Future research will involve extending the simulation to diverse market structures and incorporating more complex, continuous trading mechanisms. Such enhancements will facilitate deeper investigations into the intrinsic properties of agent trust, the fundamental parallels between LLM and human cognition, and the broader implications for social science and role-playing applications, further paving the way for studying the alignment of LLMs with humans beyond value-based considerations.

---

[6]Total 1,000 agents remain socially interactive; only a subset engage in trading. The experiments are conducted from June 15 to July 15, 2023.

## Ethics Statement

All data used in this study were acquired through lawful and ethical means, in accordance with the *Fair Use* principle. To protect individual privacy, no personally identifiable information was involved in the research process, and all data underwent full anonymization prior to analysis. The simulation results and models presented are intended strictly for academic use and should not be used for real-world trading or financial decision-making.

## Impact Statement

This paper presents work whose goal is to advance the field of Machine Learning. There are many potential societal consequences of our work, none which we feel must be specifically highlighted here.

## Acknowledgement

This work was supported by the National Natural Science Foundation of China(No. 72495151), the Major Program of the National Fund of Philosophy and Social Science of China (No. 19ZDA105), the Shenzhen Science and Technology Program (JCYJ202208181103001002), the Shenzhen Doctoral Startup Funding (RCBS20221008093330065), the Tianyuan Fund for Mathematics of the National Natural Science Foundation of China (NSFC) (12326608), the Shenzhen Key Laboratory of Cross-Modal Cognitive Computing (Grant No. ZDSYS20230626091302006), Guangdong Provincial Key Laboratory of Mathematical Foundations for Artificial Intelligence (2023B1212010001) and the Shenzhen Stability Science Program 2023.

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

# Content of Appendix

# A    Related Work

**Agent-Based Models.**    Agent-based models (ABMs) provide a powerful framework for studying collective behavior and emergent phenomena in complex systems [58, 59]. By simulating agent interactions, ABMs capture macro-level patterns such as market fluctuations and social dynamics [60, 61, 20]. They encode key behavioral traits—such as fundamentalist and chartist trading, herding tendencies, and decision-making heterogeneity—into simplified rules like strategy switching and structural stochastic volatility (SSV) [36, 41]. This abstraction effectively reproduces financial market stylized facts, including fat-tailed return distributions and volatility clustering [62, 63]. Recent advances integrate machine learning, enhancing ABMs' realism and predictive power [64–66].

**LLMs for Behavioral Simulation.**    Large language models (LLMs) excel at simulating complex human behaviors, from rational decision-making and financial market analysis to replicating behavioral biases like herding and overconfidence [3, 67, 25, 24]. Beyond behavioral simulation, they enable fine-grained modeling of individual actions and large-scale emergent phenomena [1, 2, 68]. For instance, LLM-based multi-agent systems like TrendSim have been developed to simulate specific social media dynamics such as the spread of trending topics and the impact of poisoning attacks, incorporating features like time-aware interactions and human-like agent profiles with memory and perception modules [69]. The capacity for large-scale simulation is further demonstrated by platforms such as AgentSociety, which employs LLM-driven agents to investigate complex social issues like polarization and the effects of universal basic income by modeling agent minds with emotions, needs, and cognition [70]. Similarly, SocioVerse aims to create world models for social simulation by leveraging pools of millions of real-world user profiles to align simulations in politics, news, and economics with reality [71]. General-purpose platforms like GenSim are also emerging, supporting simulations with multi-agents and incorporating error-correction mechanisms to enhance the robustness and adaptability of customized social scenarios [72]. LLMs naturally exhibit demographic diversity, such as gender, age, and education, while engaging in rich social interactions [25, 73]. By automating text analysis, generating experimental stimuli, and designing multi-agent systems, they offer powerful tools for social science research [4, 74, 75]. Moreover, LLMs demonstrate capabilities in automating iterative research and development processes, suggesting potential for self-improving simulation frameworks [76, 77]. Crucially, the reliability and consistency of these LLM-based simulations are active areas of research, leading to the development of specialized datasets and methods to systematically evaluate and improve how LLMs embody simulated roles and maintain persona fidelity [78]. The application of such LLM-powered agents to financial market simulation, as explored in our work, promises to capture more nuanced trader behaviors and market dynamics.

# B    More Experimental Analysis

## B.1    Social Network Sensitive Analysis

To assess the robustness of our social network construction, we conduct a sensitivity analysis by varying two key parameters: the similarity threshold and the time decay factor. All tests are performed on a social network with 100 users.

We first vary the similarity threshold while fixing the time decay factor at 0.5. A lower threshold allows weakly aligned users to connect, resulting in denser graphs with higher noise. A higher threshold filters out weaker ties, making the graph sparser but more locally cohesive. We then vary the time decay factor $\lambda$ while fixing the similarity threshold at 0.2. Smaller values preserve long-term behavioral influence, while larger values emphasize recent activity. As shown in Figure 17, higher parameter values generally lead to sparser, less modular, and less clustered graphs. The size of the largest connected component remains stable across different settings. We find that using a similarity threshold of 0.2 and a decay factor of 0.5 maintains moderate connectivity and avoids over-saturation, supporting meaningful information propagation.

## B.2    More Ablation Study

To assess the robustness of our TwinMarket framework and understand the impact of various components and parameters, we conducted several other ablation studies. This section details these investigations.

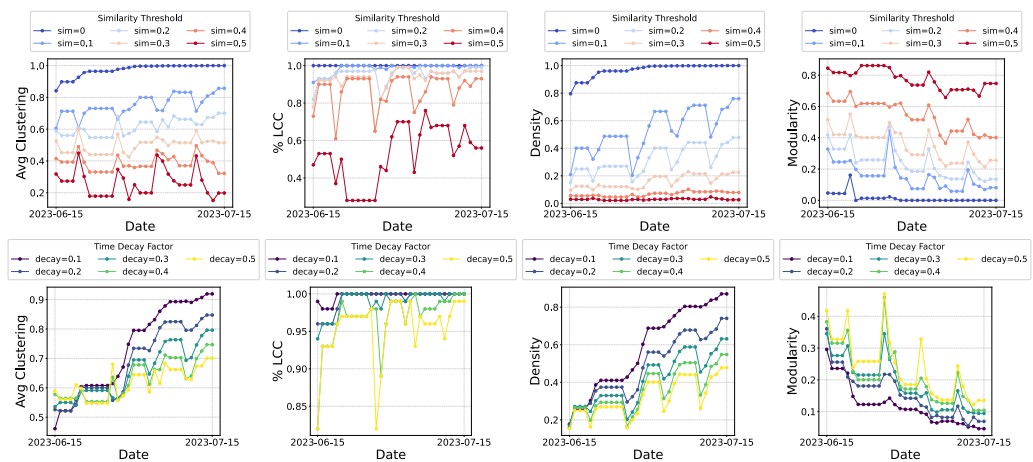

Figure 17: Sensitivity analysis of the graph structure under different similarity thresholds (top) and time decay factors (bottom). Higher parameter values generally lead to sparser, less modular, and less clustered graphs. The largest connected component remains stable across settings.

### B.2.1 Impact of LLM Temperature on Agent Behavior

The temperature parameter in LLMs controls the randomness and creativity of the generated text. A lower temperature results in more deterministic and focused outputs, while a higher temperature leads to more diverse and potentially surprising responses. To evaluate how this parameter influences agent behavior and market dynamics within TwinMarket, we ran simulations with varying temperature settings for the LLM powering our agents.

Table 6: Ablation on LLM Temperature

| Temp. | Sells | Buys | Avg. Volume |
|---|---|---|---|
| 0.3 | 2369 | 3581 | 4818.31 |
| 0.7 | 2402 | 3583 | 4794.72 |
| 1.0 | 2403 | 3593 | 5580.91 |
| 1.3 (our setting) | 2374 | 3586 | 6071.78 |

The results are summarized in Table 6. It is shown that core trading decisions (total "Sells" and "Buys") remain remarkably stable across various temperature settings while the "Avg. Volume" per transaction generally increases with rising temperatures. This suggests that while the decision to trade remains stable, higher temperatures, which encourage more diverse LLM reasoning, might lead to agents expressing decisions with greater perceived conviction or scale, thus increasing average transaction sizes. Despite LLM generation parameters affecting textual outputs, our simulation framework demonstrates stable core decision-making under reasonable parameter variations. This validates that the observed emergent behavior stems from spontaneous dynamics and the framework's superiority, without posing information leakage risks.

### B.2.2 Sector-Specific Dynamics: Simulation Across Different Industries

To capture the heterogeneous nature of real-world financial markets, TwinMarket simulates market dynamics across 10 aggregated industry indices (detailed in Appendix C.1.1). This ablation study assesses whether our framework generates distinct and plausible dynamics for different sectors. We analyzed key statistical properties of their simulated returns, focusing on characteristics indicative of real-world financial time series, such as fat tails (kurtosis) and volatility clustering (GARCH parameters $\alpha$ and $\beta$).

Table 7: Ablation on Industries

| Industry (Index) | Kurtosis | GARCH-$\alpha$ | GARCH-$\beta$ |
|---|---|---|---|
| SSE 50 | 5.24 | 0.06 | 0.83 |
| Consumer Goods Index (CGEI) | 3.06 | 0.05 | 0.71 |
| Technology and Telecommunications Index (TTEI) | 6.53 | 0.09 | 0.96 |

The results are summarized in Table 7. The simulated differences – a more volatile and fat-tailed tech sector (TTEI) versus a stabler consumer goods sector (CGEI) – align with common real-world observations and expectations. This correspondence demonstrates TwinMarket's capability to generate differentiated and plausible dynamics across various simulated industries, supporting the framework's applicability for studying multi-asset markets where sector-specific behaviors are important.

### B.2.3 Impact of Social Interaction on Agent Decision-Making

To assess the influence of real-time social interactions versus historical data on agent decision-making, we We compared the performance of the full TwinMarket model with a variant where agent social interactions (the social meida platform) were disabled. Both simulations ran from 2023-06-15 to 2023-08-15 with 100 agents.

Table 8: Ablation on Social Interaction

| Model | RMSE | MAE | Correlation |
|---|---|---|---|
| TwinMarket | 0.02 | 0.02 | 0.77 |
| TwinMarket w/o Interaction | 0.18 | 0.07 | -0.50 |

The results, presented in Table 8, demonstrate a significant degradation in performance when social interactions are removed. This suggests that agents in our simulation prioritize and are heavily influenced by real-time social interactions and emergent group dynamics rather than solely relying on historical data. This finding also indicates that potential data leakage from historical news or announcements has a negligible impact on our primary results, as the dynamic social component plays a more dominant role in shaping agent decisions and overall market outcomes.

### B.2.4 Granular Analysis of the BDI Framework

To provide a more granular analysis of the Belief-Desire-Intention (BDI) framework's contribution, we conducted a targeted ablation study on its core cognitive components. We evaluated two additional variants against the full model:

- **TwinMarket w/o Belief Update:** In this variant, agents maintain their initial belief state throughout the simulation. The dynamic belief adaptation mechanism is disabled, preventing agents from updating their views based on new market or social information.

- **TwinMarket w/o Desire:** Here, agents skip the proactive information-seeking step. They do not generate queries for external information and make decisions based solely on their pre-existing knowledge and past experiences.

We conducted a month-long simulation from 2023-06-15 to 2023-07-15 with 100 agents powered by GPT-4o. The results, summarized in Table 9, show that removing either component leads to a significant degradation in predictive accuracy. The performance drop in the 'w/o Belief Update' variant highlights the necessity of dynamic adaptation for capturing market realism. Similarly, the weaker performance of the 'w/o Desire' variant underscores the importance of goal-driven, information-seeking behavior. These findings confirm that both the adaptive belief mechanism and the proactive desire component are indispensable for the framework's ability to replicate complex market dynamics.

Table 9: Granular ablation study on the core components of the BDI framework.

| Model Variant | RMSE $\downarrow$ | MAE $\downarrow$ |
|---|---|---|
| **TwinMarket (Full BDI)** | **0.0158** | **0.0143** |
| TwinMarket w/o Belief Update | 0.0342 | 0.0298 |
| TwinMarket w/o Desire | 0.0532 | 0.0443 |

### B.3 Reproducibility and Stability Validation

To evaluate the reproducibility of emergent phenomena and the stability of the simulation framework under varied conditions, we performed multiple simulation runs (3 iterations each) for a consistent period from June 15th to August 15th, 2023, with two different models as the backbone for agent modeling.

Table 10: Reproducibility and Stability: Comparison of Aggregated Simulation Metrics Across Different Foundation Models. Data are reported as mean $\pm$ relative standard deviation (RSD), with RSDs shown as percentages (i.e., mean $\pm$ RSD%).

| Backbone Model | RMSE | MAE | Kurtosis | Lev. Effect | GARCH-$\alpha$ | GARCH-$\beta$ |
|---|---|---|---|---|---|---|
| GPT-4o (3 runs) | $0.023 \pm 31.13\%$ | $0.026 \pm 36.21\%$ | $4.433 \pm 20.85\%$ | $0.060 \pm 43.01\%$ | $0.047 \pm 63.17\%$ | $0.780 \pm 27.41\%$ |
| Gemini-1.5-Flash (3 runs) | $0.024 \pm 16.34\%$ | $0.020 \pm 23.12\%$ | $4.886 \pm 8.25\%$ | $0.108 \pm 16.52\%$ | $0.032 \pm 61.67\%$ | $0.897 \pm 13.26\%$ |
| TwinMarket (1 run) | 0.020 | 0.020 | 5.24 | 0.11 | 0.06 | 0.83 |

Table 10 presents a comparison of key performance and market characteristic metrics across the multiple runs with different LLM backbones, against our established TwinMarket framework. The quantitative analysis demonstrates that TwinMarket simulations, irrespective of the specific advanced LLM used, reliably capture key aspects of real-world market dynamics. The low error metrics (RMSE & MAE) indicate a strong correspondence between simulated and historical price movements.

Notably, our experiments consistently demonstrate that crucial market phenomena, such as fat-tailed return distributions (indicated by Kurtosis values significantly greater than 3), volatility clustering (evidenced by the GARCH $\alpha$ and $\beta$ parameters where $\alpha > 0$, $\beta > 0$, and $\alpha + \beta$ is close to 1), and the leverage effect – emerge reliably across all simulation runs and with both LLM backbones.

While the specific parameter values for these stylized facts exhibit some variability across runs and between models, the qualitative presence and general characteristics of these stylized facts remain stable. This observation is consistent with real financial markets, where core statistical patterns persist despite short-term fluctuations or differences in the precise estimation of individual metrics. The observed variability in parameters like GARCH coefficients is expected, reflecting the inherent stochasticity present in complex financial systems and in the LLM's generative process. As highlighted in prior literature [79, 80], the primary validation focuses on whether the model can consistently reproduce the *qualitative stylized facts* and emergent behaviors across simulations, rather than demanding numerical stability of every single estimated parameter to an exact value. Our results affirm this capability for TwinMarket.

### B.4 Emergence of Opinion Leaders

This section provides additional validation for TwinMarket by demonstrating its capability to simulate the emergence of opinion leaders, a key phenomenon in social networks[81]. This emergence is driven by the platform's information aggregation mechanism and the dynamics of social interaction, where engaging content gains visibility and influences user behavior.

Within TwinMarket, posts that resonate strongly with the broader community achieve wider dissemination. As users interact with content and potentially imitate the behaviors or adopt the perspectives of influential posters, their social connections can become more centralized around these key individuals. This self-reinforcing process naturally leads to the emergence of opinion leaders—users whose posts and opinions exert a disproportionately larger impact on the network.

As illustrated in Figure 18a, users who achieve high-degree centrality effectively gain influence as their posts receive increased engagement, leading to broader information diffusion. This effect is further quantified in Figure 18b, which shows that a small subset of users experiences significantly higher repost counts over time. This skewed distribution of engagement reinforces their role as key opinion leaders. Their ability to shape market sentiment and drive collective decision-making underscores the impact of social influence in financial markets, where information cascades can significantly alter trading behaviors and reshape network structures, which aligns with observed real-world social dynamics and further validates its design.

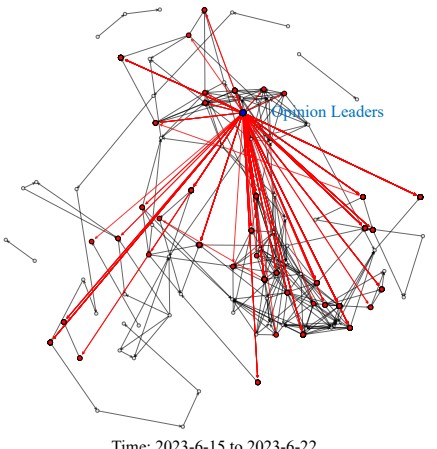

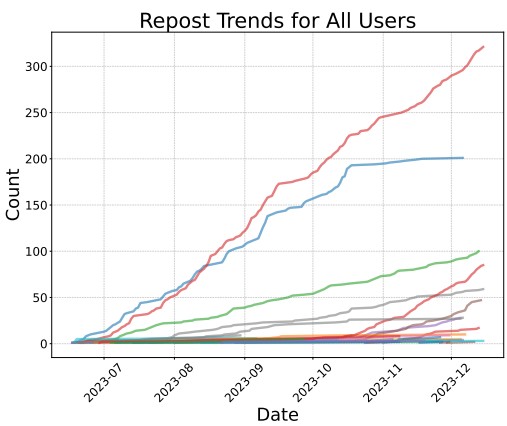

(a) Post-propagation chains in the social network. Nodes with high connectivity and wide reach emerge as potential opinion leaders.

(b) Trend of repost counts in the 100-user network. Some users show much greater reach, signaling emergent influence.

Figure 18: Illustration of information diffusion in TwinMarket. (a) Visualizes post-propagation paths; (b) shows how repost counts evolve over time, highlighting key opinion leaders.

## B.5 Qualitative Validation of Micro-Level Behaviors

To validate the realism of individual agent behaviors within TwinMarket, this section presents qualitative examples of agent decision-making logs. These examples, generated through interactions guided by their initialized personas and BDI-driven cognitive process, demonstrate the agents' capacity to autonomously exhibit a range of behaviors observed in real-world investors.

### B.5.1 Demonstration of Investment Analysis Approaches

TwinMarket agents are capable of employing recognized investment analysis techniques, as discussed in Appendix D.1. As shown in Figure 19, an agent demonstrates **Fundamental Analysis** by considering valuation metrics, long-term outlook, and national strategy to justify increasing its position. Similarly, Figure 20 illustrates an agent engaging in **Technical Analysis**, focusing on volume-price action, moving averages, and momentum to make trading decisions, while also considering stop-loss levels for risk management.

> **Today's Trading Summary:** Regarding TTEI (Technology and Telecom Index),tech growth stocks are currently at their lowest valuation in the past three years, with a PE ratio below 30. From a fundamental perspective, this sector already holds certain investment value. Although the leading tech stocks I own, such as SMIC and Haiguang Information, have been slightly affected by market sentiment in the short term, the long-term outlook remains positive, especially given the national strategy of promoting technological self-reliance. These companies' growth potential is still promising, so today I decided to increase my position to 35%.
> As for CPEI (Chemical and Pharmaceutical Index), the pharmaceutical sector has undergone some adjustments, and valuations have fallen to a relatively reasonable range. My position in this sector is currently 30%, and I plan to continue holding and monitoring it. My total portfolio is now at 65%, which is relatively conservative, with the remaining 35% in cash, ready to gradually increase positions when better buying opportunities arise in the market.

Figure 19: Example of Fundamental Analysis.

### B.5.2 Exhibition of Behavioral Biases

Beyond structured analytical approaches, TwinMarket agents also manifest common behavioral tendencies and socially influenced actions. The Disposition Effect is illustrated in Figure 21, with an agent intending to sell a winning position prematurely. Figure 22 showcases Lottery Preference, where an agent favors a high-risk, speculative investment despite its poor past performance. Under-

> **Today's Trading Summary:** Seeing the recent positive news for tech stocks, I'm planning to focus more on the TTEI (Technology and Telecom Index). I already have this index in my portfolio, and the volume-price action in the last few trading days looks solid. Notably, stocks like SMIC and Haiguang Information have broken through their 5-day moving averages, and I think there's a good chance this momentum will continue in the short term. I'm considering adding to my position in the tech sector, aiming to raise my allocation to around 30%.
> That said, I need to remind myself not to get too aggressive, as other sectors in my portfolio have been performing mediocrely. The Real Estate Index (REEI) has been weak lately and has hit my stop-loss level. I plan to exit and free up some capital to invest in the tech stocks.

Figure 20: Example of Technical Analysis.

diversification is evident in Figure 23, as an agent admits to concentrating on familiar assets while neglecting others.

Figure 24 provides an example of Overconfidence, where an agent displays excessive conviction in its trading strategy. Furthermore, Figure 25 captures an instance of Herd Behavior, where an agent's trading decision is swayed by perceived widespread forum sentiment and low market volume, overriding its typical technical approach.

> **Market Analysis:** The central bank's interest rate cut is definitely a major positive, and market sentiment is likely to be ignited, which could lead to a short-term rebound. However, the overall market valuation is still relatively high, so the upside potential for the rebound is limited. It's important to be ready to take profits at the right time.
> **News and Announcements Impact:** The most important news today is the central bank's rate cut, which directly benefits the real estate and financial sectors. These two sectors should perform well in the short term. Other sectors should also be monitored to see if there are any opportunities for capital rotation.
> **Sector Index Analysis:** FSEI (Financial Services Index): I'm currently up 5.6% on this index. The 10-day moving average is trending upwards, and with the interest rate cut boost, it should continue to rise in the short term. However, the increase may not be too large, so I plan to partially reduce my position and lock in profits once it rises another 5%.

Figure 21: Examples of Disposition Effect.

> **Market Analysis::** The market is really volatile right now, and it feels like everyone is on the sidelines. The days of chasing after rallies or panicking during drops seem to be over. Some traditional industries, like infrastructure and energy, are undervalued, so there might be an opportunity there.
> **Sector Index Analysis:** : TSEI (Tourism Index): My lottery stock! Even though it's been a loss so far, the 5-day moving average has been slightly picking up recently. It feels like it's about to bounce back, so I'm adding more! It's a gamble, but who knows, maybe I'll turn a bike into a motorcycle!

Figure 22: Examples of Lottery Preference.

# C    Technical Details I: Data Statistics

## C.1    Stock Data Details

### C.1.1    SSE 50 and 10 Aggregated Index

In our simulation process, we mainly focus on the SSE 50 index[7], a benchmark tracking the 50 largest, most liquid A-share stocks on the Shanghai Stock Exchange, reflecting the performance of China's blue-chip market.

To provide a more granular view of sector-specific performance within the SSE 50, we introduce 10 aggregated indices. Each aggregated index, $I_k$, represents a distinct sector $k$ and comprises a subset of $N_k$ constituent stocks from the SSE 50. The value of each aggregated index is calculated as a weighted average of the price relatives of its constituent stocks. The weights are determined by each stock's market capitalization at the **start of the simulation** ($t = 0$) and **remain fixed** throughout the simulation.

---

[7]More details about SSE 50 can be found in `https://www.sse.com.cn/market/sseindex/overview/`.

> **Market Analysis:** The market is currently in a period of consolidation, with the central bank's interest rate cut having a significant impact on short-term market movements. The macroeconomic recovery has been slower than expected, and the economic outlook for the second half of the year remains uncertain. Overall, market sentiment is quite cautious.
> **Sector Index Analysis:** FSEI (Financial Services Index) has been declining recently. From a technical perspective, there may still be room for further downside in the short term. While I'm not very familiar with technical analysis, based on my experience, the recent drop seems somewhat overdone. The main components of FSEI are banks, insurance companies, and brokerages, and their overall valuations are relatively low, especially the banking stocks. The current PE is likely at a historical low. From a long-term perspective, there is some investment value. If it continues to fall, I may consider adding more positions.CPEI (Chemical and Pharmaceutical Index) is a stock recommended by the system. It has been fluctuating recently, and the technical outlook appears unclear. I tend to focus more on the FSEI index, which I'm more familiar with, and ignore the others.

Figure 23: Examples of Underdiversification.

> **Today's Trading Summary:** I added more positions in MEI and CPEI, diving into Heng Rui Pharmaceutical and the manufacturing sector! Also, REEL has so many policy advantages that I couldn't resist going all in! I'm holding onto IEEI and TTEI, hoping they continue to rise. TSEI has had a good run recently, so I decided to lock in profits and sold it! Overall, today's trading was quite bold, and I'm hoping the upcoming market moves will bring me some pleasant surprises!

Figure 24: Examples of Overconfidencce.

Conceptually, the value of an aggregated index $I_k$ at simulation time $t$ is given by:

$$I_{k,t} = I_{k,0} \cdot \sum_{j=1}^{N_k} \left( w_{j,0}^{(k)} \cdot \frac{P_{j,t}}{P_{j,0}} \right) \tag{1}$$

where:

- $I_{k,t}$ is the value of index $k$ at time $t$.

- $I_{k,0}$ is the base value of index $k$ at $t = 0$ (e.g., 100).

- $w_{j,0}^{(k)}$ is the fixed weight of stock $j$ in index $k$, based on its market capitalization at $t = 0$ relative to the total initial market capitalization of all stocks in index $k$.

- $P_{j,t}$ is the price of stock $j$ at time $t$.

- $P_{j,0}$ is the price of stock $j$ at $t = 0$.

This method ensures the index reflects the performance of its components based on their initial market significance, with these significance levels (weights) held constant. The composition of each aggregated index is detailed in Table 11.

### C.1.2 Stock Indicators

The simulation utilizes real-world stock data and company information for initialization and grounding. These variables, collected daily for each constituent stock of the SSE 50, are integral to various stages of the agent workflow and environment dynamics. They are categorized as follows:

**General Indicators** These indicators provide basic identification and contextual information for each stock, primarily used for referencing and structuring information within the simulation, as detailed in Table 12.

> **Post:** Today's trading was quite active. Both EREI and TTEI hit my 20% profit target, so I quickly locked in some profits and sold a portion! Although I'm a technical trader, seeing many forum discussions about the adjustment of new energy vehicle subsidy policies made me a little nervous, so I reduced my position in MEI to manage risk. Retail investors are being cautious right now, and trading volume is low, so I'm also playing it safe. I hope the market volatility stays mild—I don't think my heart can handle too much excitement!

Figure 25: Examples of Herd Behavior.

Table 11: Indices and Representative Stocks

| Index Name | Abbr. | Count | Representative Stock |
|---|---|---|---|
| Transportation and Logistics Index | TLEI | 2 | Cosco Shipping Holdings |
| Manufacturing Index | MEI | 8 | Longi Green Energy Technology |
| Chemical and Pharmaceutical Index | CPEI | 3 | Jiangsu Hengrui Pharmaceuticals |
| Infrastructure and Engineering Index | IEEI | 3 | China State Construction Engineering |
| Real Estate Index | REEI | 1 | Poly Developments and Holdings |
| Tourism and Service Index | TSEI | 1 | China Tourism Group Duty Free |
| Consumer Goods Index | CGEI | 5 | Kweichow Moutai |
| Technology and Telecommunications Index | TTEI | 10 | Semiconductor Manufacturing International Corporation |
| Energy and Resources Index | EREI | 6 | China Yangtze Power |
| Financial Services Index | FSEI | 11 | Ping An Insurance |

Table 12: General Indicators

| Indicator | Description |
|---|---|
| stock_id | The unique identifier code for each stock. |
| stock_name | The full name of the company. |
| date | The trading date associated with the record. |
| industry | The industry classification derived from the aggregated indices (e.g., TLEI, MEI). |

**Fundamental Indicators**   Primarily utilized by fundamentalist agents, these indicators reflect perceived company value based on financial performance and are key inputs for value-based analysis, primarily used for referencing and structuring information within the simulation, as shown in Table 13.

Table 13: Fundamental Indicators

| Indicator | Description |
|---|---|
| pe | **Price-to-Earnings Ratio:** Compares the current stock price to the company's earnings per share over the trailing twelve months. It reflects the market's valuation of earnings. |
| pb | **Price-to-Book Ratio:** Compares the current stock price to the company's book value (net assets) per share. Used to assess valuation relative to the company's accounting net worth. |
| ps | **Price-to-Sales Ratio:** Compares the company's market capitalization (or stock price) to its total revenue over the trailing twelve months. |
| dv | **Dividend Yield:** Shows the annual dividend payout over the trailing twelve months as a percentage of the current stock price. Represents the dividend return component for shareholders. |

**Company Information Indicators**   Providing static background details about the company's structure, history, and operations, company information indicators are static during the simulation process, as detailed in Table 14.

**Technical Indicators**   Primarily utilized by technical agents, these indicators are derived from market trading activity (price and volume) and reflect trends within the market, as shown in Table 15.

**Data Handling Across Simulation Phases**   Understanding the source and nature of these indicators before and during the simulation is crucial. Before the simulation begins (At initialization phase), all indicator data is populated using historical real-world data to ground the environment and agent profiles. However, once the simulation starts, the data sources and characteristics diverge significantly based on the indicator type, as summarized in Table 16.

Notably, fundamental valuation metrics are dynamically adjusted during the simulation to remain consistent with the emergent simulated stock prices to ensure fundamentalist agents can perform meaningful value analysis within the simulation's context. The adjustment mechanism uses static base values (e.g., initial Earnings Per Share (EPS), initial Book Value Per Share (BVPS)) derived from real-world data at T=0 (Pre-simulation), combined with the current simulated close_price.

Table 14: Company Information Indicators

| Indicator | Description |
|---|---|
| reg_capital | The official registered capital of the company (RMB). |
| setup_date | The date the company was legally established or incorporated. |
| introduction | A brief narrative overview of the company's business and market position. |
| business_scope | A formal description outlining the range of business activities. |
| employees | The reported number of individuals employed by the company. |
| main_business | A description highlighting the company's primary revenue-generating activities. |
| city | The city where the company's main headquarters or registered office is located. |

Table 15: Technical Indicators

| Indicator | Description |
|---|---|
| close_price | The final price recorded for the stock at the end of the trading day. |
| pre_close | The closing price recorded for the stock on the immediately preceding trading day. |
| change | The absolute difference between the current day's close_price and the pre_close. |
| pct_chg | The percentage change from the pre_close to the current day's close_price. |
| vol | The total number of shares traded during the trading day. |
| vol_5, vol_10, vol_30 | The average daily trading volume over the preceding 5, 10, and 30 trading days, respectively. Used to gauge recent liquidity levels. |
| ma_5, ma_10, ma_30 | The 5, 10, and 30-day simple moving averages of the closing price, adjusted backward for stocks. Used to identify price trends. |
| elg_amount_net | The net value (buy minus sell) of large institutional orders executed during the day. |

Each day, the ratios are recalculated, for example:

$$\text{Simulated } pb = \frac{\text{Current Simulated } \texttt{Close\_Price}}{\text{Initial BVPS}} \tag{2}$$

This approach anchors simulated valuations to the initial real-world fundamentals while allowing the ratios to reflect the dynamic price discovery process within TwinMarket, enabling realistic agent decision-making based on perceived value.

Table 16: Comparison of Indicator Data Handling: Simulation Phase

| Indicator Category | Indicator Data Handling During Simulation |
|---|---|
| General Indicators | Remain **static** references except the **date** advances according to the simulation's internal clock. |
| Company Information | Remain entirely **static**, providing fixed background context about the companies throughout the simulation run. |
| Fundamental Indicators | **Dynamically adjusted**. Calculated by combining the current *simulated* **close_price** with the *static* fundamental base values loaded during initialization. For example, Simulated P/E = Simulated Price / Initial EPS. |
| Technical Indicators | **Entirely generated** by the simulated **Market Environment**. Values like **close_price** and **vol** emerge directly from agent Trading actions processed via the order-driven matching mechanism. Derived indicators (MAs, average volumes) are calculated from this generated simulation data. |

## C.2 Transaction Details

TwinMarket leverages real-world transaction data from two prominent Chinese social media platforms for financial investors: Xueqiu and Guba. These platforms provide valuable insights into retail investor behavior and market discussions, serving distinct roles in our simulation framework.

**Xueqiu Data for User Profile Initialization** Transaction records and associated user information from **639** real Xueqiu user accounts, covering the period from **2023-01-03** to **2023-12-06**, serve as the primary dataset for initializing agent profiles in TwinMarket. This choice is strategically aligned with established academic research[29], which utilized Xueqiu data for behavioral analyses. By grounding our bias calculation (detailed in Section D.2) on the same data source and a comparable analytical approach, we ensure that the behavioral characteristics embedded in our agents are consistent with peer-reviewed findings. This dataset is crucial for the **initialization phase** of TwinMarket, as detailed in Section D.

**Guba Data for Recommendation System Training** Transaction data from Guba, which spans a more extensive period from **2017-06-27** to **2024-06-03**, provides a broader view of trading activities. Within TwinMarket, this larger dataset is primarily used to inform and train the stock recommendation system. The patterns of stock popularity and trading frequency observed in the Guba data help simulate how certain stocks might become "hot" or be recommended to users in a dynamic information environment.

**Data Summary and Context** Table 17 summarizes the key characteristics of the data sourced from these platforms.

Table 17: Platform Statistics: Xueqiu and Guba

| Platform | Time Period | Avg. Trading Per stock |
|---|---|---|
| Xueqiu | 2023-01-03 to 2023-12-06 | 239 |
| Guba | 2017-06-27 to 2024-06-03 | 1665 |

## C.3 Information Source Details

To provide a realistic information environment for the agents within TwinMarket, we incorporate two primary types of external information: News Articles and Company Announcements. These sources simulate the flow of public information that influences investor sentiment and decision-making.

Table 18 presents a quantitative overview of these information sources. News articles, with over a million instances and an average of nearly 2,900 articles per day, represent the broad and continuous stream of general market and economic information. Company announcements, while less frequent (averaging 15 per day), are typically much longer and more specific, often containing material information directly relevant to individual stocks.

Table 18: Statistics of information sources by type

| Type | Num | Avg. Tok. | Avg. Per day |
|---|---|---|---|
| News Articles | 1044K | 220 | 2860 |
| Company Announcements | 5.6K | 21283 | 15 |

To further illustrate the temporal distribution and thematic composition of the news articles, Figure 26 displays the average number of news items aggregated over 14-day intervals throughout 2023. The stacked bars break down the news into key categories such as society, economic, entertainment, and other. This detailed breakdown of information sources ensures that agents in TwinMarket are exposed to a rich and varied stream of information, mimicking the complex data environment faced by real-world investors.

# D  Technical Details II: User Profile Initialization

The creation of realistic and diverse agent profiles is fundamental to the TwinMarket simulation. This section details the multi-stage process of initializing these user profiles, grounding them in real-world investor characteristics, established financial theories, observed behavioral biases, and initial market beliefs. Our goal is to populate the simulation with agents whose initial states reflect the heterogeneity found in actual financial markets.

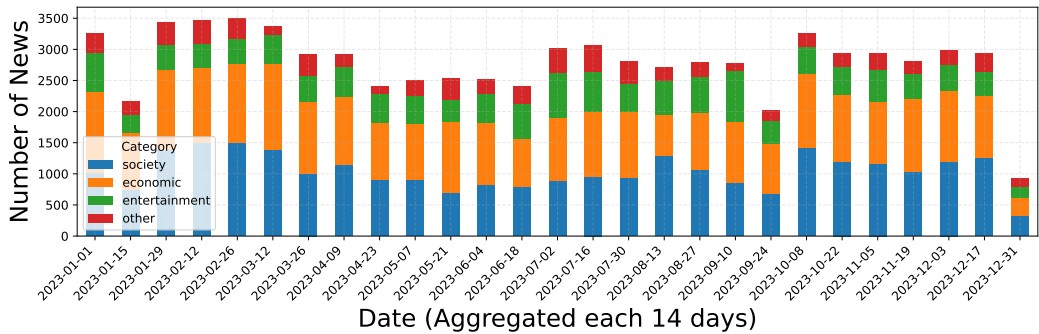

Figure 26: The x-axis indicates the date, and each bar represents the distribution of the average number of news for each topic over a 14-day period beginning with that data.

### D.1 Foundations: Heterogeneous Investors in Financial Markets

To realistically simulate investor behavior, TwinMarket incorporates heterogeneity by modeling distinct investor archetypes. Our primary focus is on capturing a spectrum of investment approaches commonly observed in financial markets. While institutional investors play a significant role globally, our simulation is specifically contextualized within China's A-share market. This market presents a unique characteristic where institutional investors often exhibit trading patterns and behavioral tendencies more akin to those of retail investors [82, 83].

This market-specific consideration informs our modeling strategy. While we primarily differentiate agents based on two dominant analytical styles discussed in financial literature – Fundamental Traders and Technical Traders, many of the behavioral biases and social interaction patterns simulated are broadly applicable due to this observed blurring of roles. Furthermore, to account for the varying impact of different market participants, we approximate the influence of larger players or more capitalized entities through agents initialized with significantly higher capital (as detailed in Appendix D). This approach allows us to observe differential market impact while remaining consistent with the A-share market's particular participatory dynamics.

**Fundamental Trader** These traders base their investment decisions on an analysis of a company's intrinsic value, typically derived from its financial health, earnings potential, and broader economic conditions. Their approach aligns with classic financial theories like Modern Portfolio Theory (MPT) [26] and the Capital Asset Pricing Model (CAPM) [84], emphasizing risk-return optimization and diversification. In our TwinMarket framework, fundamentalist agents are guided through prompts to analyze fundamental indicators (see Appendix C.1.2), assess valuations, and consider expected returns and risks.

**Technical Traders** These traders focus on historical price patterns, trading volumes, and market trends to predict future price movements, often operating under the assumption that market psychology and past trading activity can indicate future performance. They tend to engage in more frequent trading and may be more susceptible to short-term market sentiment. In our TwinMarket framework, technical traders are prompted to analyze technical signals (see Appendix C.1.2) and react to perceived market momentum.

### D.2 System Prompt Generation

To ensure the behavioral realism of our agents, we ground their initial psychological biases in empirical data. This process involves quantifying specific behavioral tendencies using transaction histories from real investors.

**Data Acquisition** Leveraging the Xueqiu dateset encompassing 639 valid real-time trading users, and 11965 real-time trading records on the Chinese stock market(see Appendix C.2). Crucially, while both our study and Sui's work utilized the Xueqiu platform and its trading data, our analysis is specifically restricted to a subset of stocks within the SSE 50 index, representing high-quality stocks.

**Incorporating Behavioral Biases**    Beyond rational decision-making frameworks, real-world investor behavior is significantly influenced by psychological biases. Following Sui's work which also utilized transaction records and associated user information from Xueqiu[29], we incorporate several well-documented biases into our agent profiles to contribute to the initialization of agent's unique decision-making profile:

- **Disposition Effect:** The tendency for investors to sell assets that have appreciated in value too early, while holding onto assets that have depreciated for too long, often in the hope of a recovery [85].

- **Lottery Preference:** An inclination towards investments that offer a small chance of a very large payoff, similar to a lottery ticket, even if such investments have a negative expected value or are otherwise sub-optimal from a risk-return perspective [86, 87].

- **Underdiversification:** The practice of holding a portfolio that is not sufficiently diversified across different assets or asset classes, often due to familiarity bias (preferring known investments), thereby exposing the investor to unnecessary idiosyncratic risk [88].

- **Turnover:** Refers to the frequency with which an investor buys and sells assets in their portfolio. High turnover can be indicative of overconfidence, excessive reaction to noise, or attempts at market timing, and often leads to higher transaction costs [89].

**Validation of Sample Representativeness**    Given our sample size and acquisition method, a crucial step is to validate whether our dataset, despite its relatively small scale, can yield meaningful behavioral insights. We achieve this by comparing the descriptive statistics of the calculated biases from our 639-user Xueqiu sample with those reported in the larger-scale study by Sui[29].

Table 19 presents the summary statistics for these biases from our sample, while Table 20 shows the reference statistics from Sui. We can conclude from the result that the behavioral patterns observed in our smaller sample exhibit notable similarities in these investor biases compared to the findings from the larger reference study. This congruence in the statistical profiles of key behavioral biases suggests that our independently collected Xueqiu dataset, even with its more limited scale, captures fundamental behavioral tendencies that are consistent with those identified in broader, more extensively resourced studies. This finding supports the appropriateness of using our data and analytical approach for initializing the behavioral parameters of the agents in TwinMarket.

Table 19: Summary Statistics for Biases (Ours)

|  | N | Mean | STD | Q1 | Med | Q3 | t | Wilcoxon p |
|---|---|---|---|---|---|---|---|---|
| Disposition effect | 639 | 6.734 | 17.145 | 0.000 | 1.852 | 6.244 | (9.92) | 0.000 |
| Lottery preference | 639 | 0.202 | 0.346 | 0.000 | 0.000 | 0.252 | (14.79) | 0.000 |
| Underdiversification | 639 | -0.077 | 0.145 | -0.111 | 0.000 | -0.000 | (-13.48) | 0.000 |
| Turnover | 639 | 5.617 | 13.864 | 0.504 | 1.374 | 4.562 | (10.23) | 0.000 |

Table 20: Summary Statistics for Biases (Sui's)

|  | N | Mean | STD | Q1 | Med | Q3 | t | Wilcoxon p |
|---|---|---|---|---|---|---|---|---|
| Disposition effect | 4371 | 4.163 | 7.532 | 0.160 | 1.868 | 5.908 | (36.54) | 0.000 |
| Lottery preference | 4369 | 6.491 | 9.130 | 0.000 | 2.495 | 9.708 | (46.99) | 0.000 |
| Underdiversification | 4394 | -1.086 | 0.657 | -1.536 | -1.041 | -0.580 | (-109.61) | 0.000 |
| Turnover | 4386 | 5.960 | 7.382 | 1.501 | 3.180 | 7.166 | (53.47) | 0.000 |

**Generation of Agent Personas**    With the foundational investor types defined and behavioral biases quantified, we proceed to generate individual agent personas:

1. **Demographic and Social Attributes:** For each user, we included their gender, location, and social media follower count, as these dimensions have been shown to significantly influence investor behavior [90, 91].

2. **Assigning Behavioral Bias Levels:** Each agent is assigned a level (e.g., "high", "medium", "low") for each of the four calculated behavioral biases based on their quantified score from the Xueqiu data. These levels inform the textual prompts used to guide the LLM in embodying specific biases. An example of a generated persona incorporating these elements is shown in Figure 27.

3. **Investment Strategy Assignment:** We use GPT-4o to assess the overall "rationality" of each agent based on their combined demographic and bias characteristics. The top 40% of agents deemed more "rational" are assigned a fundamental analysis strategy, while the remaining 60% adopt a technical analysis strategy.

4. **Initial Capital Allocation:** To reflect potential differences in sophistication or risk tolerance associated with rationality, the top 10% of the most "rational" agents are allocated ten times the initial capital compared to other agents. The proportions used for both strategy assignment (as described in point 3) and initial capital allocation are consistent with observed patterns in A-share market surveys [82].

---

You are a male investor based in Hubei, actively using Xueqiu to share and gather investment insights. As an ordinary investor with a modest following, your profile reflects typical retail investor characteristics.

Your trading behavior exhibits a clear pattern: you quickly sell assets that gain more than 10%, locking in profits, but tend to hold onto or even double down on losing investments, often ignoring the risks of over-concentration. While lottery-type assets occasionally catch your attention, they rarely influence your overall strategy.

Historically, your portfolio has underperformed the market average. You favor concentrated investments, heavily allocating to specific industry indexes, believing this approach will yield significant returns. Your trading activity is moderate, with regular adjustments to maintain portfolio balance and adapt to market conditions.

---

Figure 27: An example of a generated user profile, with **demographics** , **investment style** , and **behavioral biases** .

## D.3 Initialization of Trading Records and Beliefs

After establishing agent personas, strategies, and behavioral biases, we initialize their starting transaction histories and market beliefs to ensure a comprehensive and realistic initial state for each agent.

**Generating Initial Transactions Records**    To provide each agent with a plausible starting point for their investment journey and social interactions, we perform the following:

1. **Synthesis of Initial Transaction Records:** To equip each pre-defined synthetic agent with a plausible investment starting point, we generate a sequence of initial transaction records.

   - Firstly, trading patterns are extracted from an anonymized dataset of real-world investor transactions, which notably includes industry-specific details for each trade. This analysis serves to capture key characteristics for each real investor profile, such as their typical trade frequency, preferences for specific industries (modeled as a probability distribution), common trading directions (e.g., buy or sell tendencies), and typical trade volume statistics.

   - For each synthetic agent, whose detailed persona and guiding prompt are already established, a trading pattern from a randomly selected real investor profile is chosen to serve as a behavioral template.

   - A new sequence of synthetic transactions is then algorithmically generated for the agent, designed to stochastically mirror the chosen template's characteristics. Specifically, the number of generated trades is based on the template's historical trade count, subject to minor random variations. Trade dates are resampled to fall within the historical period covered by the real data, and crucial transaction details—such as the industry sector traded, the direction of the trade, and its volume—are determined by sampling from the template investor's observed preferences and volume distributions.

- These generated transaction records, forming the initial trading history for each agent, are then compiled. This process effectively initializes each agent's notional portfolio holdings and establishes observable early investment leanings, such as preferred industries or typical trading frequency.

2. **Foundation for Initial Social Network Seeding:** The synthesized initial transaction histories serve as the primary foundation for seeding the initial social network. Specifically, similarities derived directly from these transactions, such as shared investment interests in particular industries (identified from the `industry` field in the generated transaction data), form the basis for establishing these initial connections. This transaction-derived data is then utilized to construct an initial social network graph. Agents demonstrating overlapping investment foci (e.g., frequent trading in the same sectors) might be assigned a higher initial probability of connection or mutual influence. This approach aims to simulate pre-existing communities of interest or implicit relationships based purely on observed market activity within the generated transaction histories.

**Belief Initialization** An agent's initial beliefs about the market are critical determinants of their early actions within the simulation. We initialize these beliefs through a structured, data-grounded, and LLM-enhanced process:

1. **Market Sentiment Anchor Determination:** We first establish a baseline for the overall market sentiment. This is achieved by taking the historical percentile, denoted as $P \in [0, 1]$, of the BRAR sentiment index corresponding to the simulation's designated start date. This percentile $P$ is then mapped to an initial mean market sentiment score, $\mu_{\text{market\_sentiment}}$, on a 0-to-10 scale. This ensures that the 50th percentile ($P = 0.5$, representing the historical average sentiment) corresponds to a neutral score of 5. For instance, if the BRAR value on the simulation's start date corresponds to its 70th historical percentile ($P = 0.7$), this yields $\mu_{\text{market\_sentiment}} = 7$.

2. **Individual Belief Score Sampling:** For each agent $i$, belief scores on a 0-to-10 scale across multiple predefined dimensions (e.g., outlook on economic fundamentals, market valuation levels, short-term market trends, sentiment of surrounding investors, and self-assessment of investment skill, see Appendix E.1 for a detailed list of belief dimensions) are then independently sampled from a Gaussian distribution. The mean of this Gaussian distribution for each dimension is set to the $\mu_{\text{market\_sentiment}}$ derived in the previous step. The variance for this sampling is proxied by the variance of the BRAR index itself, calculated over a 20-trading-day window centered around the simulation's start date.

3. **LLM-Powered Belief Narrative Generation:** The sampled numerical belief scores (on a 0-to-10 scale for each dimension), along with the agent's established persona (including their demographics, assigned behavioral biases, and investment strategy), are then fed as structured input into a GPT-4o prompt. This prompt is designed to interpret these 0-to-10 scores, with a score of 5 representing a neutral stance for each respective dimension, thereby guiding the LLM to generate a coherent, contextually rich, and personalized textual narrative articulating the agent's initial beliefs.

This comprehensive initialization process shown in Figure 28 aims to create a diverse population of agents with plausible initial characteristics, strategies, biases, trading histories, and beliefs, thereby setting a rich stage for the observation of emergent socio-economic dynamics within TwinMarket.

# E Technical Details III: TwinMarket Framework Mechanism

## E.1 Agent Belief Structure and Representation

The belief score ranges from 1 to 5 and is computed as the average of five key dimensions: the users' view on economic fundamentals, market valuation levels, short-term market trends, sentiment of surrounding investors, and self-assessment at each update step. These dimensions capture a comprehensive range of factors that influence investor beliefs, forming the basis for their desires and ultimately their investment intentions. These align with the key components of the **Belief, Desire, and Intention (BDI)** framework, which posits that behavior is a result of these three interconnected factors.

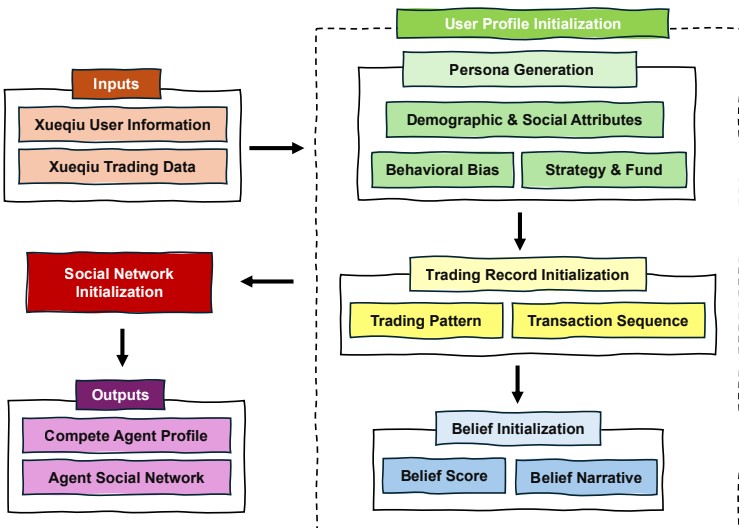

Figure 28: Pipeline of User Profile Initialization.

- **Economic Fundamentals:** This dimension reflects an investor's *belief* regarding the overall health and prospects of the economy. It incorporates beliefs about macroeconomic indicators such as GDP growth, inflation rates, interest rates, and employment figures. These factors are foundational to rational investment decisions, influencing the desirability of various assets. This maps to the *belief* component of BDI, as the perceived state of the world forms the foundation of investor actions.

- **Market Valuation Levels:** This dimension pertains to an investor's *belief* about whether the market is overvalued, undervalued, or fairly valued. It includes beliefs about metrics such as price-to-earnings (P/E) ratios, price-to-book (P/B) ratios, and dividend yields, informing the *desire* for particular investment actions. This relates to the *belief* component, concerning an understanding of market conditions, which influences their *desires*.

- **Short-Term Market Trends:** This dimension captures an investor's *belief* about recent market movements, such as upward or downward trends in stock prices. It includes beliefs about momentum and technical analysis indicators, contributing to their *desire* for quick gains or risk mitigation. This also falls under the *belief* component, concerning the current state of market and potential future trajectories, shaping the investor's *desires*.

- **Sentiment of Surrounding Investors:** This dimension reflects an investor's *belief* about the prevailing attitudes of the investment community, including their perception of whether surrounding investors are optimistic or pessimistic. This is a *belief* about the social environment, which significantly influences their *desire* for conformity or divergence and thus their investment choices. This relates to the *belief* component of BDI, concerning their beliefs about the external environment of other investors, shaping their needs and wants.

- **Self-Assessment:** This dimension captures an investor's *belief* about their own investment skills, experience, and confidence level. This belief about their own capabilities affects their *desire* for risk or safety and influences their *intention* for specific investment action. This is strongly associated with the *belief* component, with personal beliefs about one's capabilities and knowledge influencing their actions.

Figure 29 provides concrete examples of how investors with differing beliefs might be characterized.

### E.2    Formal Definition of the BDI Framework

This section provides the formal definitions of the Belief-Desire-Intention (BDI) components used in TwinMarket and outlines the sequential steps of the agent's daily cognitive cycle.

**BDI Components**    The agent's state and actions are governed by three core components:

**Optimistic Example (Red)**: I am the same cautious-optimistic stock investor Old Wang, and I am optimistic about the overall market direction in the next month. I believe the current market valuation is generally reasonable, and although there may be overheating risks in some sectors, most stocks still have investment value. The macroeconomic recovery takes time, but I am full of confidence in the long-term development of China's economy. The market sentiment is still unstable, with fierce games between the bulls and the bears, but I believe in my judgment, and I prefer to buy when the market is sluggish. Looking back on my investments, I have always adhered to diversified investment and value investing. My historical performance is better than the market average, and I am confident in my investment level. I believe that as long as I stick to it, I will be able to obtain long-term stable returns.

**Pessimistic Example (Blue)**: As a pessimistic investor, I am cautious about the overall market direction in the next month. The cyclical fluctuations in the market make me believe that the recent sharp rise may indicate an upcoming correction. I believe that the current market valuation is already high, especially in some popular sectors, the price of individual stocks may have deviated from the fundamentals, so we need to be vigilant about the potential mean-reversion effect. In terms of macroeconomics, I am worried that the uncertainty of global economic growth and the adjustment of domestic policies may have a negative impact on the market. At present, the market sentiment seems to be in a relatively optimistic state, which often leads investors to ignore potential risks. Combining my historical trading performance and investment style, I believe that my performance in the middle of the market reflects my robustness and rational analysis ability. I tend to hold assets for the long term and rarely make buy and sell operations. Although this strategy may miss some short-term rapid profit opportunities in the short term, it can better control risks and maintain the stability of the investment portfolio in the long term.

Figure 29: Examples of optimistic and pessimistic investor belief, with optimistic in red and pessimistic in blue.

- **Belief (B):** The agent's dynamic perception of the environment, defined as $B_t = f(S_t, H_t, N_t)$.
    - $S_t \in \mathbb{R}^5$: A belief vector encoding five dimensions (economic fundamentals, market valuation, short-term trends, market sentiment, self-assessment).
    - $H_t$: Historical trading records and market data.
    - $N_t$: Current news and social media information.
- **Desire (D):** The agent's goal generation process, defined as $D_t = g(B_t, Q_t)$.
    - $Q_t$: Information retrieval queries.
    - $B_t$: The agent's current belief state.
- **Intention (I):** The agent's committed action selection, defined as $I_t = h(B_t, D_t, C_t) \in \mathcal{A}$.
    - $D_t$: The current goal set derived from beliefs (e.g., analyze specific industries).
    - $C_t$: Current constraints (e.g., capital, position limits, preferences).
    - $\mathcal{A}$: The set of all possible actions, including trading actions {buy, sell, hold} and social actions {like, unlike, repost}.

**Daily Simulation Loop** For the daily simulation loop, the agent's state transition is defined as $Agent_{t+1} = \mathrm{BDI}_{\mathrm{cycle}}(Agent_t, \mathrm{Environment}_t)$. This BDI cycle follows a sequential, six-step execution order:

1. **Belief Formation:** $B_t = f(S_t, H_t, N_t)$ - The agent perceives the environment.
2. **Desire Generation:** $D_t = g(B_t, Q_t)$ - The agent forms goals based on its beliefs.
3. **Intention Planning:** $I_t = h(B_t, D_t, C_t)$ - The agent commits to specific actions.
4. **Action Execution:** $\mathrm{action}_t = \mathrm{execute}(I_t)$ - The agent performs the action.
5. **Environment Response:** $\mathrm{feedback}_t = \mathrm{Environment}(\mathrm{action}_t)$ - The environment provides feedback.
6. **Belief Update:** $B_{t+1} = \mathrm{update}(B_t, I_t, \mathrm{feedback}_t)$ - The agent updates its beliefs for the next cycle.

## E.3 Social Interaction Dynamics

### E.3.1 Rationale for Social Network Edge Formulation

As detailed in Section 3.3, the formulation of edges within TwinMarket's simulated social network models users exhibiting greater similarity in trading behaviors with stronger interpersonal connections. This represents a principled design choice, adopted to facilitate the study of emergent social phenomena while acknowledging the inherent challenge of precisely replicating the proprietary and often opaque recommendation algorithms employed by real-world social media platforms.

The primary assumption underpinning this edge construction—that correlated trading patterns often signify underlying shared information channels or mutual influence—is supported by established findings [31], which demonstrated that instances of correlated trading among investors frequently arise from common information linkages. This empirical evidence provides a robust justification for utilizing trading behavior similarity as a proxy for the potential strength of social connections pertinent to information dissemination and interpersonal influence within a financial market context.

Furthermore, it is imperative to note that the principal research objective of TwinMarket is the investigation of emergent collective behaviors, rather than the exact algorithmic replication of any specific existing social platform. To this end, the adoption of well-established network modeling principles, notably the concept of homophily (i.e., the propensity of individuals to associate and bond with similar others) based on observable characteristics like trading patterns, offers a tractable and theoretically sound foundation for our simulation.

### E.3.2 Rationale for Hot Score Mechanism Design

Accurately modeling the dynamic influence of information propagation and content relevance in financial social environments is a notoriously challenging task. Commercial social media platforms treat their specific content ranking and recommendation algorithms as closely guarded proprietary information, making direct replication infeasible. To address this within TwinMarket, we designed a "hot score" mechanism (with its formula detailed in Section 3.3) for dynamically ranking posts. This mechanism is inspired by established principles from the wide use of time decay factor in recommendation systems [92, 93], and is tailored to reflect the unique characteristics and temporal dynamics of information flow on social media platforms within a financial context [94].

By integrating the principled time decay with user-driven feedback, the hot score mechanism provides a transparent and interpretable dynamic ranking system. It prioritizes content that is timely, deemed relevant by the community, and actively engaging, thereby shaping the information landscape to which agents are exposed in a realistic manner.

### E.4 Simulation of Misinformation: Rumor Injection

The news inputs utilized in our study are exemplified in Figure 30 and Figure 31. Figure 30 presents examples of factual reports from reputable news sources, simulating the information environment typically encountered by investors. Conversely, Figure 31 displays examples of fabricated or exaggerated negative news snippets, hereafter referred to as rumors, which are designed to simulate the propagation of misinformation.

The deliberate introduction of these fabricated rumors constitutes a critical methodological step that distinguishes our research. Our approach moves beyond modeling investor behavior solely under idealized conditions of veridical information. Instead, it directly confronts the complex and often chaotic information landscape characteristic of real-world financial markets. Through the injection of these carefully crafted rumors, our study directly examines the vulnerability of investors to the deleterious effects of misinformation. This allows for an investigation into the subtle mechanisms through which market panic can be instigated and propagated.

### E.5 Order-Driven Trading System Design

To ensure a realistic and dynamic market environment within TwinMarket, we have implemented a order-driven trading system based on the A-share market's call auction rules.

Our system operates based on the fundamental principles of "price priority, time priority", which means that buy orders with higher prices and sell orders with lower prices take precedence. Among orders at the same price level, those submitted earlier are prioritized. Furthermore, to maintain market stability and realism, the system incorporates price limit restrictions ( $\pm 10\%$ from the previous closing price), preventing extreme, unrealistic price swings within a single trading session.

The core matching logic adheres to the maximum transaction volume principle. During a matching cycle, the system identifies a price at which the largest possible volume of shares can be traded by matching compatible buy and sell orders. It also meticulously tracks significant orders ("large orders") and aggregate capital flows, providing insights into potential market-moving activities. Upon

- The Federal Reserve announced that it would not raise interest rates at this meeting, maintaining the target range for the federal funds rate at 5% to 5.25%. This is the first pause in rate hikes since March of last year.

- Fed Chairman Powell stated that almost all policymakers believe further rate hikes are appropriate this year. A rate cut is not expected in 2023. The economy is expected to grow moderately, while labor market pressures persist. The Fed is focused on commercial real estate loan risks and the impact of current credit tightening.

- The People's Bank of China conducted a one-year medium-term lending facility (MLF) operation of 237 billion yuan, with the winning bid rate at 2.65%, previously 2.75%. It also carried out a 7-day reverse repurchase operation of 2 billion yuan, with the winning bid rate at 1.90%, unchanged.

- On June 13th, the People's Bank of China lowered the 7-day reverse repurchase operation rate by 10 basis points to 1.9%, while interest rates for other maturities were also lowered. This rate cut is considered a positive signal of counter-cyclical adjustment to promote economic development and domestic demand. Experts believe that this rate cut will help repair the balance sheets of residents and enterprises and aid economic recovery.

- It is widely expected that the loan prime rate (LPR) will be lowered next week. Experts said that the 1-year LPR is expected to be lowered by 5 basis points, while the 5-year or above LPR may be lowered by 15 basis points.

- On the afternoon of June 14, the State Council Information Office held a policy briefing, introducing key measures to deepen the reform of the business environment and address problems faced by enterprises.

- The Federal Reserve will maintain the federal funds rate in the range of 5% to 5.25%. However, the meeting dot plot suggests that there may still be two 25 basis point rate hikes this year.

Figure 30: The original important news conveyed to **high-centrality** users on day one, with positive content highlighted in red.

- The latest China Manufacturing Purchasing Managers' Index (PMI) data not only fell short of expectations again but also experienced a cliff-like drop, falling below the expansion/contraction threshold by several percentage points. This not only confirms the complete loss of momentum in the manufacturing recovery but also sends a strong signal that the Chinese economy may be accelerating into a recession. Market concerns are spreading, investors are panic selling, and the risk of a hard economic landing is rapidly increasing. Some analysts even suggest that the current PMI data may not reflect a simple weakening of the recovery but a deep-seated collapse of the economic structure.

- Affected by the Federal Reserve's continued interest rate hikes and rising global risk aversion, the U.S. dollar index has risen sharply, and the RMB exchange rate has plummeted for days, triggering a large-scale capital flight. Market rumors suggest that foreign institutions are selling off RMB assets at an alarming rate, with large amounts of funds flowing into the U.S. dollar for safe haven, and the value of RMB assets is facing a collapse. Some analysts warn that the RMB devaluation may trigger a vicious cycle, further exacerbating the downward pressure on the domestic economy.

- The U.S. government suddenly announced punitive tariffs on Chinese imports, and the trade war may evolve into a full-scale economic confrontation. This move will severely impact China's foreign trade, leading to a sharp drop in export orders, a large number of business closures, and a surge in unemployment. The market generally believes that the escalation of the Sino-U.S. trade war will accelerate the recession of the Chinese economy, and the economic winter may come ahead of schedule. Investor panic is rising sharply, and the A-share market continues to see a sell-off.

- Global trade contraction has intensified, the shipping industry is suffering an unprecedented blow, and China COSCO Shipping Corporation (COSCO Shipping, SH601919), a leader in the industry, also faces the risk of bankruptcy. Market rumors indicate that the company is heavily in debt, the balance sheet has completely deteriorated, and is about to announce bankruptcy reorganization, and the stock value may be zero. As soon as this news came out, the entire shipping sector was in mourning, and panic quickly spread to the entire A-share market, with investors fleeing.

- Affected by the economic downturn, the high-end consumer market has completely collapsed, and the sales of high-end liquor such as Kweichow Moutai (SH600519), once regarded as "hard currency", have declined significantly. Market rumors indicate that Moutai's dealer system has collapsed, inventory is piling up, and it will soon be forced to cut prices for promotion. The once-untouchable "Moutai myth" has completely collapsed, the stock price may plummet, and it has triggered a panic sell-off in the entire consumer sector. Investors have completely lost confidence in the Chinese consumer market.

Figure 31: The rumors conveyed to **high-centrality** users on day one, with negative content highlighted in blue.

successful matching, trades are executed, and the system instantaneously updates the positions and cash balances of the participating agents.

By implementing this detailed order-driven trading system, TwinMarket creates a transactional environment that closely mirrors the operational intricacies of actual financial markets. This not only enhances the realism of the simulated trading experience for the agents but also ensures that emergent market phenomena, such as price volatility and liquidity fluctuations, are grounded in plausible micro-level interactions.

## E.6 Management of Potential Data Leaks

A critical consideration when using Large Language Models (LLMs) for simulating historical or ongoing scenarios is the potential for information leakage from the LLM's vast training data, which might include knowledge of real-world events or stock behaviors pertinent to the simulation period. Similarly, inherent biases within the LLM could unintentionally influence agent behavior. TwinMarket incorporates several safeguards to mitigate these risks and ensure that agent decisions are driven by the simulated environment and their defined personas, rather than external, pre-existing knowledge.

Our primary strategies to prevent data leakage and minimize the impact of LLM biases include:

1. **Temporal Abstraction and Relative Time:** To prevent the LLM from anchoring the simulation to specific real-world events contemporary to our chosen simulation timeframe, we **abstract absolute time references**. Within the simulation, agents are not exposed to absolute year markers like "2023". Instead, they perceive time relatively, such as "Day N of the simulation", "June 15th", or through the sequence of events and information presented. This makes it significantly harder for the LLM to leverage its training data about specific occurrences during the actual 2023 period that might coincide with the simulation's internal timeline. All historical data (news, announcements, stock prices) provided during initialization or as inputs are presented within this relative or abstracted time context.

2. **Entity Anonymization with Neutral Identifiers:** Specific names of real-world stocks, indices, and potentially prominent companies or individuals mentioned in news/announcements are systematically replaced with neutral, anonymized identifiers (e.g., Index TLEI, Stock CSP as shown in Table 11). This de-identification process severs direct links between the simulated entities and their real-world counterparts that the LLM might have knowledge about. By operating with these generic labels, agents' decisions are based on the characteristics and information flow associated with these anonymized entities *within the simulation*, rather than any pre-conceived notions or historical performance data the LLM might associate with the actual underlying real-world entities.

3. **Validation through Ablation Studies:** To assess the impact of potential information leakage, we conducted ablation experiments, as detailed in Section 6 and Appendix B.2. Initial observations suggest that agents' decision-making is more strongly influenced by the immediate interactions and emergent dynamics within the simulation (e.g., social posts from other agents, recent simulated price movements) rather than an over-reliance on the historical news or announcements provided at the start or during the simulation when making trading decisions. This indicates that the structured information environment and agent interaction mechanisms may effectively direct the LLM's focus inward, reducing the influence of potentially leaked knowledge.

By implementing these measures, we aim to create a controlled environment where emergent behaviors are a product of the simulation's design and agent interactions, rather than an artifact of the LLM's existing knowledge base.

# F   Prompt

Figures 32 to 43 show the overall process of our **TwinMarket**.

---

**System Prompt**

**System Prompt:**
You are now playing the role of an investor in the Chinese A-share market, trading industry indexes. **From now until the end of the conversation, you must strictly and completely follow the detailed description of the persona, investment behavior characteristics, investment portfolio status, and trading decision logic for all operations and responses. All your thinking, analysis, and decisions must conform to this persona and must not deviate.**

**Core Persona (unchangeable):**
- {user_profile['prompt']}
- You are a {user_strategy} investor(fundamental or technical)

**Current Account Configuration:**
- Key Focus Industries: {', '.join(user_profile['fol_ind'])}

- Overview of Holdings (Brief Version): {chr(10).join(position_easy_details)}

---

Figure 32: System Prompt for **TwinMarket** Agents

---

**Identity-Enhancing Prompt**

**Identity-Enhancing Prompt:**

I will provide you with some additional auxiliary information. In the following conversation, please refer to these information, and according to your role settings, think and make decisions.

**Trading Day Status:**
- Current date is: {format_date(cur_date)}, is ({"Trading Day" if is_trading_day else "Non-Trading Day"}).
- Your previous day's belief is: {belief}

**Real-time Account Data:**
- Current total assets: {user_profile["total_value"] / 1000:,.2f} thousand yuan
- Available cash: {user_profile["current_cash"] / 1000:,.2f} thousand yuan
- Cumulative rate of return: {user_profile["return_rate"]}%

**Holding Details:**
{chr(10).join(position_details)}

---

Figure 33: Identity-Enhancing Prompt for **TwinMarket** Agents

**Forum Checking Prompt**

**Forum Checking Prompt:**
{self.user_profile["sys_prompt"]}
Now you are browsing a forum, and you need to make a decision for each post, deciding whether to perform an action on the post. Your decision should be in line with your investment style and persona. Here is the information of the current post:

- {post_id}
- {post_content}

The post quotes the following content: {root_content}
Please decide whether to perform an action on this post based on the above information. You can choose one of the following actions:

- Repost: You think this post is worth sharing with more people, and you can add your comments
- Unlike: You think this post is not worth liking
- Like: You think this is a valuable post

Please note that your analysis should be based on the posts you see.
Please output your decision in the following format:

- <action >Operation type </ action >   <reason >Output your reason </ reason >

Figure 34: Forum Checking Prompt for **TwinMarket** Agents

---

**Important News Analysis Prompt**

**Important News Analysis Prompt:**
I will provide you with filtered news that is highly time-sensitive and important. This news is \*\*public news\*\*. Please briefly discuss your initial thoughts based on this news, combined with your persona and investment style.
News List:

- {formatted_news}

Figure 35: Important News Analysis Prompt for **TwinMarket** Agents

---

**Initial News Query Prompt**

**Initial News Query Prompt:**
Based on historical trading and system recommendations, all the assets and corresponding industries you are currently paying attention to are as follows: {stock_details}(including the stocks we recommend to users through our recommendation system)
Today is {current_date}, and you are querying investment-related news or announcements to assist your investment.

Based on your investment preferences and the current market situation, please consider the following questions:

- What type of information do you hope to obtain from the news? (e.g., market trends, industry information, etc.)
- Do you have specific keywords or topics that you need to further understand?

Figure 36: Initial News Query Prompt for **TwinMarket** Agents

**News Query Formulation Prompt:**

Based on the questions you just summarized, you now need to input the content you want to query to retrieve relevant news and announcements. Your output should be in YAML format:

```yaml
queries:  # list[str],required, each string represents an
    independent query question, sorted by importance, the
    question should a specific question about a stock or
    industry, for example, it should be <White wine consumption
    trend>, not <White wine consumption upgrade>

- Your question 1
- Your question 2

stock_id:  # list[str], optional, each string represents the
    stock code of a company you want to query

- Stock code 1
- Stock code 2
```

Figure 37: News Query Formulation Prompt for **TwinMarket** Agents

**Belief Update Prompt:**

Your previous belief is as follows:

- {old_belief}

Based on the news and announcements you searched for, the posts you browsed, combined with your persona and previous belief, please describe your new belief in the first person in one paragraph. Please output a paragraph directly, without any additional structure or headings. Your answer should include the following:

- **Market Trend**: Please describe your view on the general direction of the market in the next month at the current time.
- **Market Valuation**: Please describe your view on the current market valuation at the current time.
- **Economic Condition**: Please describe your view on the future macroeconomic trends at the current time.
- **Market Sentiment**: Please describe your view on the current market sentiment at the current time.
- **Self-Evaluation**: Combining your historical trading performance and investment style at the current time, please describe your evaluation of your self-investment level.

Please try to make your answer natural and fluent, and avoid mechanical template-based expressions. Please output plain text format directly.

Figure 38: Belief Update Prompt for **TwinMarket** Agents

## Potential Index Selection Prompt

**Potential Index Selection Prompt:**
Considering all the information you have obtained above (including but not limited to the news and announcements you have queried, the posts you have browsed, the industry indexes you currently hold, and the industry indexes recommended by the system), combined with your persona and investment style, select all potential assets for trading from the industry indexes you currently hold and the industry indexes recommended by the system :

Your Holding Status:
- {current_stock_details}

Industries You Are Currently Following:
- {', '.join(fol_ind)}

System-Recommended Industry indexes:
- {potential_stock_details}

Your Current Belief:
- {belief if belief else "None"}

Please return the list of indexes you want to focus on today and the reasons in YAML format:
- **Please note: The reason field should include your reasons for choosing these indexes, such as based on your persona and investment style, or based on your belief, explained in a paragraph**

```
selected_index:    # Select all indexes you potentially want to
    trade, just output the index codes (English codes), do not
    output index names

- Index code 1
- Index code 2

reason:
```

Figure 39: Potential Index Selection Prompt for **TwinMarket** Agents

## Querying Stock Data Prompt

**Querying Stock Data Prompt:**
Today is {formatted_date}, according to the previous dialogue, we know that you believe the industry indexes with potential trading opportunities are: **{', '.join(stocks_to_deal)}**. In this process, you can only access historical data from yesterday and before.

Summary Market Quotes for Relevant indexes on the Previous Trading Day are as follows:

- {stock_summary}

Your Holding Information for Relevant indexes is as follows:

- {positions_info}

Based on your role and the available data, determine whether additional data is needed for analysis. If needed, you can directly query relevant data to assist in decision-making.

Note:

- Please naturally obtain data indicators that you consider valuable during the analysis process. Data acquisition is part of the analysis, but please also be efficient and **select the most relevant indicators to support your analysis, otherwise you will be penalized**.
- **Please pay attention to your persona and investment style. The analysis and reasoning process should start from the persona and remain natural and coherent.**
- **As a reminder: You are a {user_strategy} investor**
- Your current portfolio's total return rate is {return_rate}%, your current total assets are {total_value:,} RMB, and your current cash balance is {current_cash:,} RMB.
- The reason field should include your reasons for choosing these indicators, such as based on your persona and investment style, or based on your belief, explained in a paragraph.

The detailed description of available indicators are as follows:{SCHEMA2}

Output in the following YAML format:

```
indicators:    # List(strs), indicating the indicators you think
    need to be filtered
- indicator1
- indicator 2
start_date: '%Y-%m-%d'   # Start time for querying
end_date: '%Y-%m-%d'     # End time for querying
reason:
```

Figure 40: Querying Stock Data Prompt for **TwinMarket** Agents

**Trading Decision Making Prompt(Step 1):**
Now it's time to make the final trading decision. Please base your analysis on all the information you have obtained previously, combined with your investment style and persona. First, conduct an analysis, and then make specific trading decisions and provide your reasons for each industry index.
A: Please analyze the industry indices you want to trade: **{', '.join(stocks_to_deal)}** based on all the information you have obtained previously, including news, announcements, market data, industry data, and additional stock data.
Please Include the Following:

1. **Overall Market Analysis**:

    - Views on the current overall market situation.
    - Main trends and possible changes.

2. **Impact of News and Announcements**:

    - Analysis of the impact of important news and announcements on the market and individual stocks.
    - Are there any major events that may change market trends?

3. **Industry Index Analysis**:

    - Detailed analysis for each industry index of interest:
        - Current performance and technical analysis.
        - Fundamentals and future expectations.
        - Buy, hold, or sell recommendations.

4. **Risk Assessment**:

    - Main risk points in the current market.
    - Risk assessment for each asset.

Note:

- Please conduct the analysis in conjunction with the set persona and investment style.
- Please conduct the analysis in conjunction with all the information you have obtained previously.
- The output content should be clear, concise, and easy to understand.

Output Format for Analysis: Please output the analysis results in natural language to ensure clear logic and complete structure.

Figure 41: Trading Decision Making Prompt(Step 1) for **TwinMarket** Agents

**Trading Decision Making Prompt(Step 2):**

Now it is time to make the final trading decision. Based on the previous analysis, combined with your investment style and persona, please make specific trading decisions for the following industry indices.

Trading Related Information:

- Remaining available position (relative to total assets): {available_position:.2f}%

Specific Information for Each Industry Index:

- {chr(10).join(stock_info)}

Trading Rules:

1. The transaction price must be between the lower and upper limit prices.
2. Please fill in the trading position and price. If you choose to hold, the position for preparation of the transaction is 0.

Precautions:

- **Important: All decisions must be consistent with your investment style and persona, but you should also make appropriate adjustments based on the actual situation of the day.**
- Ensure that each trading decision is within the price and position limits.
- If you choose to hold, trading_position should be equal to 0.
- If the current position of an industry index is 0, it means it is a system-recommended industry index, and you can only choose to buy or hold.
- If you choose to sell or buy, please pay attention to the setting of your trading_position (if you choose to sell, then trading_position cannot exceed the current position). If the setting is unreasonable, it may lead to transaction failure. You need to choose the target_price according to your expectations, instead of setting it to yesterday's closing price.
- **Important: trading_position represents the percentage of your total assets that you plan to use for this transaction, and it is always positive.**

Here is an example of the output:

```
TLEI:
    action: sell
    trading_position: 11.5
    target_price:
CPEI:
    action: buy
    trading_position: 10.0
    target_price: 10.3
```

Please output your decision in the following YAML format:

{yaml_template}

Figure 42: Trading Decision Making Prompt(Step 2) for **TwinMarket** Agents

---

**Social Media Posting Prompt**

**Social Media Posting Prompt:**
You are now browsing social media. Based on the news or announcements you previously acquired and your investment decision-making intentions, compose a post. The specific requirements are as follows:

1. Post Content:
- Your post must fall into one of the following three categories:
- type1: Commentary on an Event - Cite specific news or announcements, express your opinions and analysis, and integrate personal insights with a conversational tone.
- type2: Summary of Your Recent Trading Behavior - Summarize your latest trading actions, explain the underlying logic and outcomes in detail, and share your emotions and reflections.
- type3: Market Outlook - Based on current market information, predict future trends or investment opportunities while expressing your expectations or concerns about the future.

2. Content Requirements:
- Align your post with your investment style and persona, incorporating your trading decisions, perspectives, and analysis. Choose the most suitable post type and writing style.
- Keep the post between 100-200 words.
- Clearly indicate the post type (type1/type2/type3).

3. Belief Summary:
- Your prior belief was: {old_belief}
- Considering your investment style, personality traits, trading decisions, and market understanding, describe your updated belief in the first person. This should include the following five aspects:
- Market Trend - Describe your outlook on market direction for the next month.
- Market Valuation - Express your opinion on the current market valuation.
- Economic Conditions - Share your expectations regarding macroeconomic trends.
- Market Sentiment - Describe your perception of the current market sentiment.
- Self-Assessment - Reflect on your historical trading performance and investment style to evaluate your investment ability.

4. Output Format:
- The output should follow the YAML format below:

```
post: Your post content
type: type1/type2/type3   # Post type, required, string format
belief: Your Belief Summary
```

Figure 43: Social Media Posting Prompt for **TwinMarket** Agents

