# OpenReview forum: "TwinMarket: A Scalable Behavioral and Social Simulation for Financial Markets"
_NeurIPS.cc/2025/Conference — NeurIPS 2025 poster_

### Official Review · Reviewer_8JKm · 2025-06-29

**Clarity:** 1
**Significance:** 4
**Originality:** 4
**Rating:** 4
**Confidence:** 5

**Summary:**

The paper attempts to create a financial network agent-based model. The paper evaluates macro-level economic behavior based on a belief-desire-intention agent framework. This framework takes in information about news and the current portfolio and makes trading decisions. This is evaluated by a number of financial metrics.

**Questions:**

See Strengths and Weaknesses.

**Ethical Concerns:**

["NO or VERY MINOR ethics concerns only"]

**Final Justification:**

- **Novelty and Significance:**
  The paper opens a promising new direction by modeling stock-market trading with agent-based methods, representing a solid first step in applying ABM to financial markets. The approach is timely and relevant for both finance and machine learning communities.

- **Behavioral Realism and Evaluation:**
  The framework aims to faithfully reproduce human-like trading behavior, not just optimize financial performance. It encodes and evaluates key behavioral finance metrics (disposition effect, lottery preference, underdiversification, turnover), showing that simulated agents exhibit realistic and diverse trading behaviors grounded in empirical data.

- **Limitations and Future Directions:**
  While the macro-level results are convincing, the micro-level mechanics (order execution, price formation) are simplified, and the zero-sum environment is a limitation. However, these choices are justified for computational tractability and benchmarking, and the authors have clear plans to address these in future work, including richer market dynamics and more realistic trading mechanisms.

**Limitations:**

The mentioned limitation is the trading scenarios, but as mentioned in strengths and weaknesses, the micro analysis of the agents behavior leaves much to be desired.

**Quality:**

2

**Strengths And Weaknesses:**

The paper opens an intriguing new direction by modeling stock-market trading with agent-based methods. Its premise is timely and warrants a series of deeper studies, and this manuscript represents a solid first step. Although the experimental analysis is framed for a financial audience, readers in machine learning would benefit from moving key implementation details now confined to Appendices F and G into the main text. Understanding how the agents are constructed and validated is just as important as the macro-level economic trends highlighted below.

In section 4.1 evaluation: Every empirical finance study should report standard baselines so that performance is interpretable. In particular, please include the buy-and-hold return on a broad index (e.g., S&P 500) and compare each agent’s return against it whenever results are discussed. The text also needs to clarify whether your claim is that TwinMarket (i) reproduces human-like trading behavior or (ii) merely produces strong financial performance. If the former, what evidence demonstrates fidelity to real traders?

Is this a zero-sum trading environment? Then some agents need to do very poorly for others to get rich. If the agents are substantiated similarly, this may not be possible. The only evidence of the difference is the return for each agent. There needs to be more specific analysis of the belief and traits, maybe what you are looking for with desires and intentions is a utility function? This is not clearly defined.

While the macro events convincingly illustrate information diffusion, the micro-level mechanics of order execution and price formation are evaluated only superficially. It is unclear if the instances of the agents with belief and intent are actually mimicking specific trader preferences or only show the macro market behavior when grouped together.

---

> ### Author Rebuttal · Authors · 2025-07-31
>
> **Dear Reviewer 8JKm,**
>
> **Thank you for taking the time to review our paper and for your constructive comments. We appreciate your careful evaluation and the concerns you raised. Below, we address your specific comments in more detail:**
>
> > Question 1: Key implementation details currently in Appendices F and G should be moved into the main text for clarity and accessibility, especially for machine learning readers.
>
> In the main text, we prioritized providing an essential financial background to help ML readers understand the context of the simulation. To maintain narrative flow and readability, we moved some of the more detailed technical and implementation aspects to Appendices F and G. We agree that bringing more of these details into the main body could enhance clarity, and we will incorporate key implementation information in the next version to improve accessibility for ML readers.
>
> > Question 2: In Section 4.1, standard baselines (e.g., buy-and-hold return on S&P 500) should be reported to contextualize agent performance.
>
> We agree this will greatly enhance the clarity of our results. In the next version, we will add figures showing (i) the return distribution of all agents, and (ii) a direct comparison of the cumulative returns of the simulated market index, all agents, and the real-world A50 index.
>
> > Question 3: The paper should clarify its main claim — whether it aims to (i) reproduce human-like trading behavior or (ii) generate strong financial performance. If (i), what evidence supports the behavioral realism of the agents?
>
> Our main claim is (i): **TwinMarket aims to faithfully reproduce human-like trading behavior**, rather than to optimize for strong financial performance. To support the behavioral realism of our agents, we carefully selected four widely recognized behavioral finance metrics [1] to construct agent prompts: **disposition effect, lottery preference, underdiversification, and turnover** (see Appendix F.2 for details).
>
> To rigorously evaluate the behavioral realism of TwinMarket agents, we compute four key behavioral metrics and test whether they are significantly different from zero, following [2]. The corresponding statistics from Xueqiu real-world data are also reported for reference:
>
> |**Behavioral Bias**|**Xueqiu (Real Data)**|**TwinMarket (Simulation)**|**Wilcoxon p-value**|
> |-|-|-|-|
> |**Disposition Effect**|6.734|3.120|0.000|
> |**Lottery Preference**|0.202|0.025|0.082|
> |**Underdiversification**|-0.077|-0.802|0.000|
> |**Turnover**|5.617|5.045|0.000|
>
> Except for lottery preference (which is significant at the 0.1 level), the other behavioral indicators are significantly different from zero at the 0.01 level. These results demonstrate that our framework faithfully captures the key behavioral biases present in real-world investors.
>
> We acknowledge that there are still some quantitative discrepancies between the simulated agents and the real Xueqiu users. Addressing these gaps is an important direction for our future work.
>
> [1] Barberis, N., & Thaler, R. (2003). A survey of behavioral finance. Handbook of the Economics of Finance, 1, 1053-1128.\
> [2] Sui, P., & Wang, B. (2023). Social transmission bias: Evidence from an online investor platform. Available at SSRN 4081644.
>
> > Question 4: In a zero-sum environment, return differences among agents require explanation. The paper lacks analysis of how variations in beliefs, traits, desires, or intentions (e.g., utility functions) contribute to profitability.
>
> Our simulation operates in a zero-sum trading environment with no exogenous capital inflow or outflow. The observed differences in agent returns therefore arise purely from interactions among agents with heterogeneous behavioral patterns.
>
> To ensure diversity in agent behaviors, we explicitly encode a range of behavioral traits into agent prompts, resulting in a spectrum of decision-making styles. To directly analyze how these individual differences relate to profitability, we compute the correlation matrix between four key behavioral indicators and final returns. The results are presented below. The `p-values` of the correlation coefficients are reported in parentheses:
>
> ||disposition_effect|lottery_preference|underdiversification|turnover|return|
> |-|-|-|-|-|-|
> |disposition_effect|1|||||
> |lottery_preference|0.227 (0.023)|1||||
> |underdiversification|0.081 (0.424)| 0.081 (0.422)|1|||
> |turnover| 0.138 (0.171)|0.052 (0.606)| -0.088 (0.382)|1||
> |return| -0.118 (0.05)|-0.077 (0.000)|0.084 (0.406)|-0.120 (0.001)|1|
>
> We observe that, except for underdiversification, the other behavioral biases: **disposition effect, lottery preference, and turnover** are all negatively correlated with final returns, with several relationships reaching statistical significance.This is consistent with findings in behavioral finance: irrational, emotion- or price-driven behaviors (often with delayed reactions) typically lead to wealth losses, while agents whose decisions are more grounded in fundamentals are more likely to capture gains from others.
>
> These results provide direct evidence that the diversity in beliefs, traits, and intentions among our agents leads to meaningful and interpretable differences in profitability, thus faithfully reflecting the mechanism of wealth redistribution in real financial markets.
>
> > Question 5: The evaluation of micro-level mechanisms (order execution and price formation) is shallow. It is unclear whether agents reflect specific trading preferences or merely produce aggregate market patterns when grouped.
>
> In the current version of TwinMarket, order execution is implemented as a single call auction at the end of each trading day, which is a deliberate simplification to ensure computational efficiency for large-scale, long-horizon simulations. We acknowledge that this abstraction limits the resolution of intraday trading and microstructure effects. Extending the framework to support multiple intraday auctions or continuous trading mechanisms is an important direction for our future work, as it will allow for more realistic modeling of order flow and price formation at a finer time scale.
>
> Regarding whether agents merely generate aggregate market patterns or genuinely reflect specific trading preferences: our framework explicitly encodes heterogeneous behavioral traits, such as disposition effect, lottery preference, underdiversification, and turnover, into each agent's persona via their prompts.We observe that variation in these traits leads to meaningful differences in agent-level outcomes, including profitability and trading style (see correlation analysis in response to Question 4). Thus, TwinMarket agents not only collectively generate realistic macro-level market phenomena, but also individually exhibit diverse, interpretable trading behaviors grounded in behavioral finance theory and empirical data.
>
> **Best regards,**
>
> **Authors**

---

> > ### Comment · Reviewer_8JKm · 2025-08-06
> >
> > I appreciate the clarification for Q1 and Q2. Q3 I acknowledge the financial behavior as adequate. Thank you for the clarifications.
> >
> > Q4 and Q5. I still find the zero-sum trading environment as slightly questionable, but I am confident the authors will continue to strengthen and relax assumptions in future iterations of their work. I strongly believe that more ABM should be accepted as strong contributions to the Machine Learning literature and urge reviewers to acknowledge the contributions of ABM as a method that will be useful for the general ML community.

---

> > > ### Author Response · Authors · 2025-08-06
> > >
> > > **Dear Reviewer 8JKm,**
> > >
> > > Thank you very much for your thoughtful follow-up and supportive remarks regarding our work. Regarding your comments on Q4 and Q5, we would like to further clarify our methodological choices and outline our future directions.
> > >
> > > Both the assumption of a zero-sum trading environment and the use of a single daily clearing price are widely adopted in traditional ABM of financial markets [1]. These simplifications are commonly used to ensure computational tractability, facilitate the analysis of wealth redistribution mechanisms, and isolate the effects of agent heterogeneity and interaction on market outcomes. By adopting these settings, our primary goal was to enable rigorous, equivalent comparison with established ABM benchmarks and to focus on the emergent macro-level phenomena arising from micro-level agent interactions. Comparing with traditional ABM is also one of our core motivations, as we aim to address research gaps in modeling richer agent behaviors and their impact on market dynamics [2].
> > >
> > > However, we acknowledge that these assumptions also have limitations. The zero-sum constraint, for example, omits capital inflows and outflows that are prevalent in real markets, potentially limiting the realism of wealth dynamics and long-term market evolution. Similarly, restricting order execution to a single daily auction abstracts away important features of intraday price formation and microstructure effects.
> > >
> > > In future directions, we plan to address these limitations by integrating our framework into a broader socio-economic simulation, allowing investors to dynamically deposit and withdraw funds from the financial subsystem. This enhancement will enable modeling of capital flows and investor behavior beyond the zero-sum constraint, supporting richer and more realistic market dynamics. We also intend to extend our framework to support continuous trading mechanisms (e.g., multiple daily auctions or a limit order book), which will enhance the realism of our market microstructure modeling and open up new applications in high-frequency trading research and market stress testing.
> > >
> > > Thank you again for your constructive feedback and for recognizing the value of ABM in advancing ML research. We are committed to further strengthening our work along these lines.
> > >
> > > [1] Axtell, Robert L., and J. Doyne Farmer. "Agent-based modeling in economics and finance: Past, present, and future." Journal of Economic Literature 63.1 (2025): 197-287.
> > >
> > > [2] Gürcan, Önder. "Llm-augmented agent-based modelling for social simulations: Challenges and opportunities." HHAI 2024: Hybrid human AI systems for the social good (2024): 134-144.
> > >
> > > **Best regards,**
> > >
> > > **Authors**

---

### Official Review · Reviewer_rR9u · 2025-07-02

**Clarity:** 4
**Significance:** 3
**Originality:** 3
**Rating:** 4
**Confidence:** 4

**Summary:**

This paper proposes TwinMarket, a multi-agent framework that leverages LLMs to simulate investor behavior in a stock market. It features a design driven by the belief-desire-intention framework. Experiments are performed to simulation a 5-month period in Mainland China stock market, demonstrating that LLM agents can effectively model real-world behaviors, validate behavioral theories, and reveal underlying mechanisms of social emergence.

**Questions:**

Q1. Do you have results on the distribution of (1) the number of stocks in the portfolio, (2) assets, and (3) positions, across the 100 agents?

Q2. For predictive accuracy, RMSE, MAE, and correlation are measure on what aspect? Stock price?

Q3. In Figure 11(c) and Figure 13, I wonder how the transactions made by the 100/1000 agents affect the index price. In the real world, it is unlikely that the transactions made by 100/1000 retail investors have a strong impact on the index price. Institutions may cause such an impact. But I don't think they are BDI-driven, and the underlying factors of their behavior are more complex than individuals.

Q4. Based on my knowledge, a trading lot equals 100 shares in Mainland China. That is, when buying stocks, the quantity must be in multiples of 100 shares. Are the agents aware of this rule in the simulation?

Q5. Is cash a dynamic value? For example, investors may get salary paid every month and use it in the market.

Q6. For information propagation, do opinion leaders have stronger trading power (i.e., more assets) and use such trading power to affect the market? While this may contradict my opinion in Q3, I'm just curious about the setting in the experiments.

**Ethical Concerns:**

["NO or VERY MINOR ethics concerns only"]

**Final Justification:**

The authors' rebuttal addressed some of my concerns, especially for the experimental settings and measures. However, the simulated scenario in this paper only covers a 5-month span in the A-share market of mainland China, which lacks shorting and hedging, hence excluding it from a clear accept. This work could benefit from simulating more markets (e.g., NYSE, with more variety of transactions) and longer periods (e.g., full bull-bear cycles).

**Limitations:**

Yes.

**Quality:**

3

**Strengths And Weaknesses:**

S1. Using LLM agents to simulate human behavior is a promising direction and has received considerable attention recently.

S2. The experiments are extensive, covering both micro- and macro- factors. Ablation studies are sufficient, and multiple LLMs are tested as well.

S3. The paper is well-written and easy to read.

W1. The simulation only covers a scenario of SSE50 starting from June 15, 2023 for five months. This is a bear market with relatively low volumes. It is encouraged that more scenarios are also evaluated to better understand the performance of the proposed framework. For example, bull markets and black swan events, which are rare (financial crisis, etc.) but have significant impacts on the market.

W2. It is unclear whether the framework can model long-horizon behavior, especially for the transition of investment style. For example, an investor may be a day trader with very frequent transactions at the beginning and gradually become a long-term trader. This is a common practice for many investors, because in the long run, we don't have enough time to keep an eye on the market. Risk tolerance and resistance to rumors may also change over time. Moreover, even veterans may change their investment style on rare events, e.g., they tend to exhibit speculative behavior under bullish policies (like late Sep. 2024 in China stock market).

W3. It is unknown whether the framework supports shorting and hedging. As far as I know, retail investors are not allowed to short stocks in Mainland China. So I suppose shorting was not implemented in the simulation. There are other ways of hedging besides shorting stocks, but I didn't find them in the paper either.

W4. While the paper claims it scales up to 1000 agents, most experiments are conducted on 100 agents. In addition, some experimental settings are unclear (please see questions below), especially for the reason why the transactions made by 100/1000 agents (as retail investors, I suppose) will significantly affect the stock price.

---

> ### Author Rebuttal · Authors · 2025-07-31
>
> **Dear Reviewer rR9u,**
>
> **Thank you for taking the time to review our paper and for your constructive comments. We appreciate your careful evaluation and the concerns you raised. Below, we address your specific comments in more detail:**
>
> > W1: the simulation lacks scenario diversity
>
> Our research goal is to validate a framework that links individual behavior to macro-level market phenomena. We began with a relatively stable period, which served as a clean testbed for assessing whether the model can reproduce the key **stylized facts** of real markets (Sec. 4.2). This controlled environment let us isolate the model's intrinsic dynamics without confounding shocks.
>
> We then extended the analysis to more turbulent scenarios. By actively inducing optimistic beliefs, the simulator generated an endogenous boom-bust cycle (Sec. 5.1, Fig. 8), and by injecting targeted negative rumors, it reproduced a **black-swan-type crash** (Sec. 5.2). Both experiments produced realistic emergent behaviors, including spiraling price swings and belief polarization, demonstrating the framework's ability to capture high-impact dynamics. Applying TwinMarket to longer historical spans that cover full bull-bear cycles and major crises remains an important direction for future work.
>
> > W2: modeling long-horizon behavioral shifts and investment style transitions
>
> We thank the reviewer for raising the question of whether our framework can model long-horizon investment behavior and style transitions. Our design explicitly supports the simulation and analysis of long-term behavioral evolution.
>
> First, our agent's behavior is anchored in a multi-dimensional Belief state that captures the key factors underlying long-term investment behavior: economic fundamentals, market valuation levels, short-term market trends, the sentiment of surrounding investors, and self-confidence (Sec. 3.2, Appendix G.1). These belief components jointly determine risk tolerance, position sizing, and openness to rumors—core aspects that drive both short-term and long-term investor behavior.
>
> Second, our framework can **endogenously reproduce both short-term fluctuations and long-term style transitions**. As illustrated in Section 5.2, even veteran agents whose baseline persona is typically stable temporarily shifted to speculative trading in response to a rumor shock, demonstrating that short-term information can override long-held strategies. Similarly, in a prolonged bull market, the same mechanism would allow for increased risk-taking among normally conservative investors.
>
> Most importantly, **the belief update cadence is fully configurable**. While our current experiments use daily updates to capture microstructure and rapid dynamics, it is straightforward to switch to monthly, quarterly, or annual updates for long-horizon studies. Because each agent's belief trajectory is explicitly recorded, researchers can trace not only day-to-day reactions but also gradual, endogenous changes in investment style, such as a transition from day trading to long-term holding as circumstances evolve.
>
> In summary, our framework provides a flexible and theoretically grounded platform for simulating both short-term responses and long-term behavioral shifts in investment style.
>
> > W3: It is unknown whether the framework supports shorting and hedging.
>
> Thank you for raising this important question. Our framework does not incorporate shorting or hedging mechanisms. Due to the lack of an effective short-selling mechanism in the A-share market[1], the effectiveness of hedging strategies is reduced. As a result, mainstream agent-based modeling (ABM) studies of the Chinese market typically do not consider shorting or hedging strategies.
>
> [1] Ji, J., et al. (2018). Modifying a simple agent-based model to disentangle the microstructure of Chinese and US stock markets. Quantitative Finance, 18(12), 2067-2083.
>
> > W4: some experimental settings are unclear
>
> For all the valiation experiments and ablation experiments, we run on 100 agents. In Sec. 7 we scale up to 1,000 agents to demonstrate that our data-driven framework not only maintains performance at larger scales but also benefits from additional resources, suggesting its potential for enhanced fidelity and richer dynamics in future studies of large-scale agent system.
>
> > Q1: Do you have results on the distribution of (1) the number of stocks in the portfolio, (2) assets, and (3) positions, across the 100 agents?
>
> Yes, we tracked these distributions to ensure behavioral realism. To clarify, agents in our simulation trade 10 aggregated industry-level indexes, not individual stocks.
>
> For initial assets, we mirrored real-world A-share market data, where surveys indicate that the top 10% of investors contribute over 50% of total market liquidity. As detailed in Appendix F.2, we assigned 10x initial capital to the top 10% of agents (selected for rationality assessed by GPT-4o) to reflect this empirical distribution.
>
> During the simulation, agents on average held 2.23 **distinct industry indexes** and maintained a **position** of 60.38% of their assets, consistent with distributions observed among real retail investors. The detailed distributions are summarized below:
>
> |Industry Indexes Held|Agents (%)|
> |-|-|
> |1 industry index|28|
> |2 industry indexes|44|
> |3 industry indexes|16|
> |4 industry indexes|7|
> |≥ 5 industry indexes|5|
> |**Avg. # of Indexes Held**|**2.23**|
>
> |Position Sizing (%)|Agents (%)|
> |-|-|
> |0-25|5|
> |25-50|33|
> |50-75|44|
> |75-100|18|
> |**Avg. Position** |**60.38%**|
>
> > Q2: For predictive accuracy, RMSE, MAE, and correlation are measure on what aspect? Stock price?
>
> RMSE, MAE, and correlation are computed on the **time‑aligned daily normalized index price series**, comparing our Simulated Market Index Price (TwinMarket output) with the corresponding Historical Index Price (the groud truth real-world price) over the simulation window. RMSE/MAE are errors on normalized index levels, and correlation is the Pearson correlation between the two level series.
>
> > Q3: In Figure 11(c) and Figure 13, I wonder how the transactions made by the 100/1000 agents affect the index price. In the real world, it is unlikely that the transactions made by 100/1000 retail investors have a strong impact on the index price. Institutions may cause such an impact. But I don't think they are BDI-driven, and the underlying factors of their behavior are more complex than individuals.
>
> While institutional dominance exists, our focus is on A-share market, where "institutional retailization" [2, 3] blurs lines between institutions and retail. We approximate large players via high-capital agents (Appendix C.3), aligning with this market's unique dynamics.
>
> [2] Tan, L., et al. (2023). Retail and institutional investor trading behaviors: evidence from China. Annual Review of Financial Economics, 16.\
> [3] Jones, C. M., et al. (2025). Retail trading and return predictability in China. Journal of Financial and Quantitative Analysis, 60(1), 68-104.
>
> > Q4: Based on my knowledge, a trading lot equals 100 shares in Mainland China. That is, when buying stocks, the quantity must be in multiples of 100 shares. Are the agents aware of this rule in the simulation?
>
> Yes, the 100-share lot rule is strictly enforced in our simulation for all buy orders.
>
> > Q5: Is cash a dynamic value? For example, investors may get salary paid every month and use it in the market.
>
> Yes, cash is a core dynamic state variable, but its dynamics arise exclusively from executed trades. Concretely, an agent's cash balance is updated only when orders fill: it decreases on buys and increases on sells. Price movements of existing holdings change portfolio value, not cash. The evolving cash balance then acts as a hard budget constraint that conditions subsequent position sizing and actions.
>
> To maintain experimental control and isolate the effects of information flow and social interaction, the current study does not model exogenous cash inflows/outflows (e.g., monthly salaries or deposits). For longer‑horizon settings, incorporating exogenous income is a natural extension; we will note this as future work.
>
> > Q6: For information propagation, do opinion leaders have stronger trading power (i.e., more assets) and use such trading power to affect the market? While this may contradict my opinion in Q3, I'm just curious about the setting in the experiments.
>
> In our framework, an agent's system prompt (Appendix F.2) includes **demographic attributes** (e.g., gender, location, follower count), **quantified behavioral biases** (disposition effect, lottery preference, underdiversification, turnover), and **an assigned investment strategy** (fundamental or technical) based on a rationality assessment. These factors jointly shape the agent's personality and behavior. Influence emerges dynamically through interactions and is not directly tied to asset levels or trading capacity.
>
> Importantly, the emergence of opinion leaders is not pre-assigned or based on fixed attributes like wealth. Instead, it arises stochastically from dynamic social interactions. We observe that influential agents can emerge regardless of their initial asset levels or trading activity, indicating that opinion leadership in our framework is decoupled from financial dominance and driven more by interaction patterns and belief dynamics.
>
> **Best regards,**
>
> **Authors**

---

> > ### Comment · Reviewer_rR9u · 2025-08-02
> > **Re: Authors Rebuttal**
> >
> > Thanks for the authors' effort to respond to the review comments. They successfully addressed some of my concerns. However, seeing that the simulated scenario only covers a 5-month span in the A-share market of mainland China, which lacks shorting and hedging, I will keep my ratings.
> >
> > While I understand this outcome may be a little disappointing, my overall evaluation is still positive, and I wish all the best to the authors for the evaluation made by the AC.

---

> > > ### Author Response · Authors · 2025-08-05
> > >
> > > **Dear Reviewer rR9u,**
> > >
> > > Thank you for your thoughtful feedback and for acknowledging our efforts in the rebuttal. We understand your remaining point regarding the scope of our simulation. Our primary objective with this study was to validate our framework's ability to **observe how macro-level social phenomena emerge from micro-level agent behaviors,** even within a controlled and stable market period. We believe this validation is a key contribution in itself.
> > >
> > > We agree that extending our analysis to more complex scenarios, including markets with shorting and hedging as you suggested, is an important direction for future exploration. Thank you again for your constructive evaluation and positive overall assessment.
> > >
> > > **Best regards,**
> > >
> > > **Authors**

---

> > > > ### Comment · Reviewer_rR9u · 2025-08-05
> > > > **Re: Official Comment by Authors**
> > > >
> > > > I understand that simulations on more markets may cost considerable time and effort. The conclusion can be made stronger by integrating those simulations. That is, your findings are not pertaining to a five-month period in a specific market but can be observed in a broader range.
> > > >
> > > > Nonetheless, considering the quality of your rebuttal, I would be happy to see this paper eventually gets accepted. Besides, it seems I'm the only reviewer who have responded to your rebuttal. Let's try to get them ENGAGED!

---

> > > > > ### Author Response · Authors · 2025-08-09
> > > > >
> > > > > Thank you for your continued support and constructive feedback. We appreciate your understanding about the computational demands of additional market simulations and agree that broader validation would strengthen our findings.
> > > > >
> > > > >  We are also grateful for your positive assessment and efforts to engage other reviewers.
> > > > >
> > > > > We truly appreciate your support and will thoughtfully incorporate all suggested revisions.
> > > > >
> > > > > **Best regards,**
> > > > >
> > > > > **Authors**

---

### Official Review · Reviewer_Dupq · 2025-07-03

**Clarity:** 3
**Significance:** 2
**Originality:** 3
**Rating:** 4
**Confidence:** 4

**Summary:**

This paper introduces a novel multi-agent framework called TwinMarket that uses a group of LLM agents to simulate the trading and message posting in a socio-economic environment. The simulation is grounded in the belief-desire-intention (BDI) framework, and partially utilizes the real-world trading data collected from various sources.

**Questions:**

- Some key design components lack sufficient clarity. For example, is a belief represented as a sentiment score (1-5) in the prompt? How does each agent update its belief? How are desires and intentions represented? What input data is used for predicting the next action? What is the trading intensity used for?
- Some figures are hard to interpret. For example, there are still different peaks between the red and blue curves in Fig. 7. How does the mismatch tell us? Another example is Fig. 3. What's the definition of wealth share? How is it computed from the observational data? I understand that these are financial indices/metrics, but having a reference or a short description would improve the readability.
- The hot score is computed in a heuristic way. How reliable is this score in recommending related posts?

**Ethical Concerns:**

["Major Concern: Data privacy, copyright, and consent", "Major Concern: Data quality and representativeness"]

**Final Justification:**

I've read the authors' responses. Most of the concerns are addressed.

**Quality:**

3

**Strengths And Weaknesses:**

Strengths:
- This paper presents a very interesting experiment using LLM agents to role-play different human individuals in a trading system as well as on a social media platform. The simulation reveals findings that align well with existing micro-/macro-level economic phenomena observed from real-world data.
- The simulation has been analyzed in multiple facets, including comparison with traditional ABM baselines, analysis of how individual agents’ behavior contributes to a macro-level phenomenon, and ablation/scalability of the designed components.
- If the codebase is released, it'd be a valuable resource for social science researchers.

Weakness:
- The simulation is only compared with traditional ABMs. Additionally, the ablation study only compared variants w/o BDI and without heterogeneity. However, it remains unclear how sensitive the simulation is to other design components (e.g., belief update strategy, graph structure, neighbor similarity estimation).
- The experiments are prioritized to analyze the socioeconomic phenomenon of the simulation (e.g., rumor propagation, bias polarization). This work is another successful demo of LLM agents in social simulation. The findings may be valuable for social science researchers; however, it remains unclear when and why LLM agents may succeed or fail in such a simulation, and how the framework generalizes to other simulation applications. In other words, the intellectual merits of this work to the ML community are unclear. How can we translate this finding to develop better role-playing agents or better training/test-time algorithms?
- Another concern is data privacy. The user profiles are collected from several stock forums. However, the potential data privacy, copyright, or ethical issues are not clearly discussed. It's unclear whether it's illegal to use this data.

---

> ### Author Rebuttal · Authors · 2025-07-31
>
> **Dear Reviewer Dupq,**
>
> **Thank you for taking the time to review our paper and for your constructive comments. We appreciate your careful evaluation and the concerns you raised. Below, we address your specific comments in more detail:**
>
> > Weakness 1.1: only compared with traditional ABMs
>
> Our work is compared primarily to traditional ABMs because our core contribution is a novel framework that enhances the ABM paradigm itself. We achieve this by replacing simplistic, rule-based agents with sophisticated LLM agents capable of complex reasoning. As we believe this is the **first** use of interactive LLM agents for stock market simulation, a comparison with the previous generation of models, namely traditional ABMs, provides the most logical benchmark for our inaugural work.
>
> > Weakness 1.2: lack of sensitivity analysis for other key components
>
> In Fig. 15 and Appendix D.1, we analyze two key hyperparameters: `similarity threshold (s)` and `time decay factor (t)`, which govern graph construction (line 155) and temporal weighting of trades (line 150) from the perspective of graph. The `s` avoids overly sparse graphs, ensuring meaningful information propagation, while the `t` emphasizes recent trading behavior. Both are designed to preserve **agent heterogeneity**, which is also a central goal of our belief update strategy.
>
> To further analyze the sensitivity of the system to these components, we repeated the simulation (100 agents, gpt-4o) for the period from 2023-06-15 to 2023-07-15, and the results are summarized in the two tables below. As `t` decreases and `s` increases, the graph density rises significantly, which leads to stronger homogeneity in social network interactions.
>
> We calculated (1) **the avg number of user interactions per post**, and (2) **the avg interaction interval time** during this simulation period. The results show that as graph density increases, each post interacts with users more frequently, and the interaction intervals shorten. This demonstrates a trend of environmental information becoming more homogeneous, which explains why excessive graph density reduces the diversity of agent decisions and, consequently, dampens the dynamics (RMSE, MAE) of the simulated market.
>
> |`t` (with `s` = 0.2)|RMSE|MAE|Avg. # of Interactions|Avg. Interval Time (days)|Density (07-15)|
> |-|-|-|-|-|-|
> |0.1|0.0379|0.0336|10.96|3.04|0.93|
> |0.3|0.0548|0.0477|11.64|4.27|0.62|
> |0.5 (our settings)|0.0158|0.0143|10.34|4.54|0.51|
>
> |`s` (with `t` = 0.5)|RMSE|MAE|Avg. # of Interactions|Avg. Interval Time (days)|Density (07-15)|
> |-|-|-|-|-|-|
> |0|0.0300|0.0286|29.59|0.10|1|
> |0.1|0.0173| 0.0155|10.80|3.98|0.78|
> |0.2 (our settings)|0.0158|0.0143|10.34|4.54|0.51|
>
> > Weakness 2: contribution to the ML community
>
> Although we aim to address research gaps in social science through LLM agents, our framework also offers valuable insights for the ML community. Recently, how to enable LLM to better simulate and understand human interactions has received considerable attention [1, 2]. In our work, we demonstrate that LLM agents can more accurately simulate human behavior compared to traditional rule-based ABMs. We attribute this success to our designed BDI cognitive simulation and social interaction with real-world data. We also analyze this success and discuss the contribution of each compoents in the ablation study in Sec. 6 and Appendix D.2.
>
> Furthermore, recent RL research already shows that decision-making policies improve when trained on rich, human-like interaction logs [3]; yet such data may be scarce and costly to collect. As our LLM agents generate realistic traces that mirror observed market behaviour, the same framework can furnish high-quality synthetic data to bootstrap or fine-tune role-playing agents whenever real logs are limited, providing practical contributions to ML training pipelines.
>
> [1] Strachan, James WA, et al. (2024). Testing theory of mind in large language models and humans. Nature Human Behaviour 8.7, 1285-1295.\
> [2] Akata, Elif, et al. (2025). Playing repeated games with large language models. Nature Human Behaviour: 1-11.\
> [3] Zhang et al. (2025). Shop-R1: Rewarding LLMs to Simulate Human Behavior in Online Shopping via Reinforcement Learning. arXiv:2507.17842.
>
> > Weakness 3 & Ethical Concerns
>
> **1. Data Privacy, Copyright, and Consent**
>
> We appreciate the emphasis on privacy, copyright, and consent, and we designed our study with a privacy‑by‑design philosophy:
> - **We model behavioral patterns, not specific user:** As detailed in Appendix E & F, we use **anonymized, public data** to estimate population-level statistical distributions of behavioral traits. These aggregated statistics, not individual records, are then used to **seed entirely new, synthetic agent personas**. These agents are **fictional** and do not correspond to any real individual, consistent with established academic pracitce on social media [4].
> - **Technical Safeguards Against Data Leakage:**  All data used for calibration were **PII‑scrubbed** (e.g., removal of usernames). Inside the simulator, we (1) **abstract absolute time** to a relative timeline and (2) **neutralize real‑world entities** so agent decisions depend on in‑simulation signals, not external knowledge.
>
> **2. Data Quality and Representativeness**
>
> As detailed in Appendix F.2, we compare our sample's distributions of key behavioral biaseswith a larger, same‑source benchmark study [4].  The results show that the two distributions closely align, indicating that despite its modest size, our sample **is behaviorally representative of the target investor population**, giving us confidence that our findings are grounded in realistic behavior.
>
> [4] Sui, P., & Wang, B. (2023). Stakes and investor behaviors. SSRN Working Paper, 4219861.
>
> > Question 1.1: Is belief represented through sentiment scores (1–5) in the prompt?
>
> Yes. This score is an avg of five sentiment scores in these dimensions: **Economic Fundamentals, Market Valuation Levels, Short-Term Market Trends, Sentiment of Surrounding Investors, Self-Assessment**.
>
> > Question 1.2: How does each agent update its belief?
>
> The belief update is a dynamic, daily process that concludes the agent's cognitive loop. As described in Sec. 3.2, after completing all actions for the day, each agent evaluates its behavior based on both its actions and the environmental feedback (social media, trading signals) it received. The mechanism for this is Fig. 37, which explicitly asks the agent to synthesize all the day's information.
>
> > Question 1.3: How are desires and intentions represented?
>
> In the original BDI theory [5], Desires represent goal states, forming a list of objectives the agent wishes to achieve. Intentions are those desires the agent has committed to, stored as a structured set of goals and associated plans for action. In our framework, we adapt these concepts to the financial context: a `Desire` refers to the agent's information-seeking behavior, such as generating queries to retrieve stock or news data, while an `Intention` corresponds to the agent's final, committed action after deliberation, such as executing a trade or posting on social media, as described in Sec. 3.2.
>
> [5] Bratman, M. (1987). Intention, plans, and practical reason.
>
> > Question 1.4: What input data is used for predicting the next action?
>
> The agent's next action is based on a combination of its internal state and perceived external information. Key inputs include its persona, strategy, and current belief (Fig. 31–32), browsed social media posts (Fig. 33), queried news and announcements (Fig. 35–36), system-recommended stocks and holdings (Fig. 38), and requested historical market data (Fig. 39). These are explicitly defined in the prompt at each stage.
>
> > Question 1.5: What is the precise role of the trading intensity metric?
>
> Trading intensity is a time‑decayed activity weight per user–industry pair used solely to build the dynamic social network: we compute weighted Jaccard similarity over these vectors to set edge weights (who is similar to whom). This network then powers information aggregation and the hot‑post recommendation shown to each agent.
>
> > Question 2.1: In Fig. 7, red and blue curves still exhibit mismatched peaks — what does this imply?
>
> Fig. 7 is intended to verify that the simulator reproduces `volatility clustering`, meaning large price changes are followed by further large changes and small ones by small changes. Perfect alignment of daily peaks is neither expected nor required, because the model is not designed for point-by-point prediction of market paths. The peak mismatch therefore reflects the intrinsic randomness of financial markets while still preserving the correct persistence pattern, which is the focus of this test.
>
> > Question 2.2: In Fig. 3, how is wealth share defined and computed from observational data?
>
> In Fig. 3, `wealth share` denotes the fraction of total assets held by a specified subgroup of agents. For example, the `Top 10 % Wealth` line is computed by summing the portfolio values of the richest ten percent of agents and dividing by the aggregate assets of the entire population. Displaying this metric alongside the Gini coefficient shows that our framework can generate realistic wealth divergence from micro-level interactions. We will add a concise definition to the manuscript for clarity.
>
> > Question 3: Hot score design
>
> The hot score is a heuristic inspired by Facebook's EdgeRank [6], combining neighbor similarity (as affinity), vote difference (as engagement), and time decay. We calibrate the decay so a post's influence lasts about two weeks. While not optimized for accuracy, it effectively captures realistic exposure and social influence in our simulation.
>
> [6] Birkbak, A., & Carlsen, H. (2016). The world of Edgerank: Rhetorical justifications of Facebook's News Feed algorithm. Computational Culture (5), Special Issue on Rhetoric and Computation.
>
> **Best regards,**
>
> **Authors**

---

> > ### Author Response · Authors · 2025-08-05
> > **Looking forward to further discussion**
> >
> > **Dear Reviewer Dupq,**
> >
> > We thank you for your valuable comments and constructive feedback. We appreciate your recognition of our multi-agent simulation framework. We have addressed your concerns about our work in detail and provided additional results to answer your questions. For your concern about **ethical issues**, we have also provided a detailed response to the ethics reviewer. We sincerely hope that our responses will help clarify the contributions of our work.
> >
> > We are sincerely grateful for the effort you have invested in helping us improve our work. With the discussion period deadline approaching, we would greatly appreciate any further comments you might have. Thank you once again for your thoughtful feedback and support!
> >
> > **Best regards,**
> >
> > **Authors**

---

> > ### Comment · Reviewer_Dupq · 2025-08-06
> >
> > Thank you for your detailed responses. Most of my concerns have been addressed. I have a few follow-up questions.
> >
> > - Copyright issue (not privacy): Is it legal to scrape the webpage data without the consent of the users and website managers?
> > - Additional ablation study: Is it feasible to do an ablation study to drop one of the BDI components (i.e., belief, desire, and intent)? Additionally, to improve clarity, the draft would benefit from adding a formal definition of the three components.
> > - While it's potentially useful for creating synthetic training data using this framework, it'd be good to add a discussion about data quality control.

---

> > > ### Author Response · Authors · 2025-08-07
> > > **Copyright Issues**
> > >
> > > **Dear Reviewer Dupq,**
> > >
> > > Thank you for raising the important question regarding copyright and data usage. We will address this in the next version of the manuscript. In our research, we collected the following publicly available data:
> > >
> > > - User information from *Xueqiu* data, covering the period from 2023-01-03 to 2023-12-06, used solely for initializing agent demographics.
> > > - Trading records from *Xueqiu* data, spanning 2023-01-03 to 2023-05-26, used to initialize agent trading histories, initial positions, and investment strategies for simulations starting in June 2023.
> > > - Transaction data from guba data, from 2017-06-27 to 2024-06-03, used exclusively for training our recommendation system.
> > >
> > > All of the above data were publicly displayed on the respective websites, voluntarily disclosed by users who chose to publish their trading records on their profiles. We did not access any private, password-protected, or otherwise non-public content, nor did we employ any automated tools that would bypass terms-of-service restrictions.
> > >
> > > We are aware that *Xueqiu* updated its `robots.txt` on August 2, 2024 to explicitly prohibit the use of its content for AI system development. However, our data collection was completed prior to this update. At the time of collection, neither `robots.txt` nor the published terms of service explicitly forbade the use of publicly displayed user information for academic research. Regarding *Guba*, its data have always been publicly accessible and have never imposed restrictions on academic use.
> > >
> > > Furthermore, our approach follows the doctrine of **"fair use"** for academic research, which permits the use of public data for non-commercial, transformative purposes. Our work is fully consistent with this definition: all data used were sourced from **publicly accessible** web pages, and our objective was not to republish or redistribute content, but to create a fundamentally new scientific work by analyzing statistical properties of aggregate behaviors. This methodology is widely recognized and accepted in academic research, and aligns with precedents set by top-tier finance journals [1-2], both of which have used *Xueqiu* data for scholarly studies and have publicly released their datasets for academic purposes [3-4].
> > >
> > > In summary, our approach is well-grounded in legal principle, academic precedent, our research methodology, and the temporal context of our actions. In accordance with these principles and our responsibility to data sources, we have committed to not releasing any raw data. Any future open-sourced artifacts will consist solely of synthetic, non-reversible data.
> > >
> > > [1] Sui, Pengfei, and Baolian Wang. "Stakes and investor behaviors." Journal of Financial Economics 172 (2025): 104146. \
> > > [2] Sui, Pengfei, and Baolian Wang. "Social transmission bias: Evidence from an online investor platform." Available at SSRN 4081644 (2023). \
> > > [3] Sui, Pengfei; Wang, Baolian (2025), "Stakes and Investor Behaviors_Pseudo Data and Replication Code", Mendeley Data, V2, doi: 10.17632/2hzby3mr89.2 \
> > > [4] Sui, Pengfei; Wang, Baolian (2025), "Social Transmission Bias", Mendeley Data, V1, doi: 10.17632/wcz9hbpdhj.1
> > >
> > > **Best regards,**
> > >
> > > **Authors**

---

> > > > ### Author Response · Authors · 2025-08-07
> > > > **Synthetic Data**
> > > >
> > > > **Dear Reviewer Dupq,**
> > > >
> > > > Thank you for this valuable suggestion regarding data quality control for synthetic datasets generated by our framework. We agree that ensuring the reliability and representativeness of synthetic data is essential for its utility in ML training pipelines.
> > > >
> > > > In our framework, we address data quality in several ways:
> > > >
> > > > 1. All agent behaviors are calibrated using real-world distributions of key behavioral biases and trading patterns, ensuring the synthetic traces faithfully mirror empirical data;
> > > > 2. We systematically validate that emergent macro- and micro-level phenomena in the simulated data align with established market statistics and behavioral metrics;
> > > > 3. For additional rigor, we plan to implement further filtering and post-processing procedures, such as outlier detection and behavioral consistency checks, to enhance the fidelity of released synthetic datasets.
> > > >
> > > > We believe these measures will help ensure that the synthetic data produced by TwinMarket meets high standards of realism and quality, thereby supporting its use for robust ML model development. Besides, for your previous concern about the potential contribution, our work addresses the fundamental limitations of traditional ABM in behavioral complexity and adaptability, which will be useful for the general ML community.
> > > >
> > > > **Best regards,**
> > > >
> > > > **Authors**

---

> ### Author Response · Authors · 2025-08-07
> **Additional Ablation Study: BDI Framework**
>
> **Dear Reviewer Dupq,**
>
> For more clarity, we provide the following formal definitions of the three BDI components:
>
> 1. **Belief (B)**: The agent's dynamic state perception: $B_t = f(S_t, H_t, N_t)$
>
>     where:
>     - $S_t \in \mathbb{R}^5$: belief vector encoding five dimensions [economic fundamentals, market valuation, short-term trends, market sentiment, self-assessment]
>     - $H_t$: historical trading records and market data
>     - $N_t$: current news and social media information
>
> 3. **Desire (D)**: The agent's goal generation process: $D_t = g(B_t, Q_t)$
>
>     where:
>     - $Q_t$: information retrieval queries
>     - $B_t$: current beliefs state
>
> 3. **Intention (I)**: The agent's committed action selection: $I_t = h(B_t, D_t, C_t) \in \mathcal{A}$
>
>     where:
>     - $D_t$: current goal set derived from beliefs (e.g., analyze specific industries, evaluate stocks)
>     - $C_t$: current constraints (capital, position limits, preferences)
>     - $\mathcal{A}$ includes trading actions {buy, sell, hold} and social actions {like, unlike, repost}
>
> For the daily simulation loop: $\mathrm{Agent}_{t+1} = \mathrm{BDI\_cycle}(\mathrm{Agent}_t, \mathrm{Environment}_t)$, the BDI cycle follows a sequential execution order:
> > 1. **Belief Formation**: $B_t = f(S_t, H_t, N_t)$ - agent perceives environment
> > 2. **Desire Generation**: $D_t = g(B_t, Q_t)$ - agent forms goals based on beliefs
> > 3. **Intention Planning**: $I_t = h(B_t, D_t, C_t)$ - agent commits to specific actions
> > 4. **Action Execution**: $\text{action}_t = \text{execute}(I_t)$ - agent performs actions
> > 5. **Environment Response**: $\text{feedback}_t = \text{Environment}(\text{action}_t)$ - environment provides feedback
> > 6. **Belief Update**: $B_{t+1} = \mathrm{update}(B_t, I_t, \mathrm{feedback}_t)$ - agent updates beliefs for next cycle
>
> The three BDI components are highly interdependent in our framework. As shown in the sequential execution order above, each component builds upon the previous one. Removing the `intention` component would break this pipeline and make it impossible to observe meaningful outcomes, since `intention` directly guide the agent's subsequent actions. (In the ablation study w/o BDI in Sec. 6, we illustrate this by letting the agent directly output trading decisions.) Therefore, we propose the following targeted ablation experiments that preserve the framework's functionality while isolating each component's contribution:
>
> 1. **w/o Belief Update**: Agents maintain static beliefs throughout the simulation. All agents retain their initial belief state across all simulation dates, eliminating the dynamic belief adaptation mechanism ($B_{t+1} = B_t$ for all $t$).
>
> 2. **w/o Desire**: Agents skip proactive information seeking. During simulation, agents do not actively query external information ($Q_t = \emptyset$) and make decisions based solely on their past experiences and pre-existing knowledge.
>
> To evaluate these ablation variants, we conducted a month-long simulation from 2023-06-15 to 2023-07-15 (100 agents, gpt-4o). The results are summarized below:
>
> |Method|RMSE|MAE|
> |-|-|-|
> |w/o Belief Update|0.0342|0.0298|
> |w/o Desire|0.0532|0.0443|
> |Full BDI|0.0158|0.0143|
>
> These results (lower is better) indicate that both `belief` and `desire` play crucial roles in enabling adaptive and purposeful agent behavior, as removing either leads to significant performance degradation.
>
> **Best regards,**
>
> **Authors**

---

> > ### Comment · Reviewer_Dupq · 2025-08-07
> >
> > Thanks. If all the above responses can be integrated into the final draft, I'm fine to increase my score. I'll defer the judgment of the data issue to the ethical reviewers.

---

> > > ### Author Response · Authors · 2025-08-07
> > >
> > > Thank you once again for your kind and constructive feedback. We truly appreciate your support and will make sure to thoughtfully incorporate all the suggested revisions into the next version of the manuscript.
> > >
> > > **Best regards,**
> > >
> > > **Authors**

---

### Official Review · Reviewer_4qQH · 2025-07-06

**Clarity:** 3
**Significance:** 3
**Originality:** 4
**Rating:** 5
**Confidence:** 4

**Summary:**

The paper presents TwinMarket, a multi-agent simulation framework using LLM-driven agents embedded in a BDI (Belief-Desire-Intention) architecture to model investor behavior and social interactions in financial markets. The authors claim the approach better captures behavioral biases and emergent market phenomena like bubbles and volatility clustering, validated via stylized facts and ablation studies.

**Questions:**

**Ungrounded Assumption:** In section 3.3, is it realistic to assume users with similar trading patterns are more likely to interact? Is there any reference to ground this assumption? Smilarly for time-weighted trading intensity, is this kind of modeling from BDI literature? If yes, a proper referencce is needed.

**Missing discussion:** [1] also uses BDI for agent modeling. It worth a discussion in the main paper.

[1] Can large language model agents simulate human trust behavior?

**Quantitative benchmarking:** The authors show stylized fact metrics (e.g., kurtosis, leverage effect coefficients) approximate real market data. Did the authors perform any hypothesis tests or confidence interval analyses comparing simulated and real distributions?

**Ethical Concerns:**

["NO or VERY MINOR ethics concerns only"]

**Limitations:**

yes

**Paper Formatting Concerns:**

Typos: “Transanctions” in Figure 1

**Quality:**

3

**Strengths And Weaknesses:**

**Strengths:**

**Actually quite ambitious:** the framework does a good job incorporating social interactions and heterogeneity in agents.

**Impressively extensive experiments:** micro-level (wealth inequality, belief updates) and macro-level validations (stylized facts) are thorough and cover a wide range of real-world phenomena. The ablation studies are solid; they clearly demonstrate the necessity of both BDI modeling and agent heterogeneity. Scalability is well-explored, scaling up to 1,000 agents with plausible dynamics.

**Weaknesses:**

**The real world gap:** The simulation still lacks real-time adaptive feedback from genuine external data. It's effectively a closed system, making it questionable for studying real markets.

**Quantitative benchmarking:** The authors show stylized fact metrics (e.g., kurtosis, leverage effect coefficients) approximate real market data. Did the authors perform any hypothesis tests or confidence interval analyses comparing simulated and real distributions?

---

> ### Author Rebuttal · Authors · 2025-07-31
>
> **Dear Reviewer 4qQH,**
>
> **Thank you for your thoughtful and encouraging review. We sincerely appreciate your positive assessment of our work and your recognition of its potential. Below, we address your specific comments in more detail:**
>
> > Weakness 1: the real-world gap
>
> TwinMarket is designed as an open, data-driven system that addresses this concern in two key ways.
>
> First, our agents' behaviors are not based on arbitrary rules but are empirically calibrated using **real-world data**. As detailed in Appendix F.2 and Tables 18 & 19, we initialize core behavioral biases (e.g., disposition effect, lottery preference) by quantifying them from a dataset of **11,965 real-world investor transactions**. This grounding ensures our agents' initial decision-making frameworks are statistically aligned with observed human patterns.
>
> Second, the simulation is not a closed loop; it is dynamically driven by a continuous stream of external events. We inject over **a million real news articles and thousands of company announcements** (Table 17), prompting agents to continuously adapt their beliefs and actions in response to real-world information. This design combines initial empirical calibration with subsequent event-driven adaptation, creating a realistic behavioral laboratory for studying how information shapes emergent phenomena (e.g., our rumor propagation experiment in Sec. 5.2), which differs from the goal of real-time price forecasting.
>
> > Question 1: ungrounded assumption
>
> **1. Social network interaction**
>
> Our assumption that users with similar trading patterns are more likely to interact is supported by empirical findings in finance. Colla & Mele [1] demonstrate that correlated trading often stems from shared information linkages, suggesting that similarity in trading behavior can serve as a proxy for indirect communication or common influences. Similar approaches have also been adopted in recommender systems research [2], where user behavior is used to construct interaction networks. Since direct social networks are typically unavailable in financial data, modeling interactions based on behavioral similarity is both practical and empirically justified.
>
> [1] Colla, P., & Mele, A. (2010). Information linkages and correlated trading. Review of Financial Studies, 23(1), 203–246.\
> [2] Xia, Lianghao, et al. "Multi-behavior graph neural networks for recommender system." IEEE Transactions on Neural Networks and Learning Systems 35.4 (2022): 5473-5487.
>
> **2. Time-weighted trading intensity**
>
> The intensity function does not borrowed from classical BDI theory, its adoption is supported by three complementary strands of literature:
>
> - **Market Microstructure:** In high‑frequency order‑flow modelling, each trade is treated as a self‑exciting event in a Hawkes process, whose intensity is $\lambda(t)=\mu+\sum_{k} e^{-\beta\,(t-t_k)}$ , exactly mirroring our weight $\sum_{t} e^{-\lambda \Delta t}$ (line 150).
> Extensive empirical studies show that such exponential kernels capture the rapid decay of a trade's influence on subsequent activity [3].
> - **Recency Effects in Decision Making:** The bias that human attention and memory overweight recent information has been documented both in psychology and in investor behaviour studies [4]. Our decay term operationalises the same principle, giving higher weights to the most recent trades that best reflect current intentions.
> - **Real‑time BDI implementations:** Although the kernel itself is external to BDI, real‑time extensions of BDI agents regulate action strength according to remaining time budgets [5]. We adopt that scheduling idea but employ a smooth exponential decay rather than a hard deadline, aligning network dynamics with the agents' internal timing mechanism.
>
> [3] Bacry et al. (2015). Hawkes processes in finance. Market Microstructure and Liquidity, 1(01), 1550005.\
> [4] Mohrschladt, H. (2018). The impact of recency effects on stock market prices. In 31st Australasian Finance and Banking Conference.\
> [5] Traldi, A., et al. "Real-Time BDI Agents: A Model and Its Implementation." IJCAI. International Joint Conferences on Artificial Intelligence, 2022.
>
> > Question 2: missing discussion
>
> We thank the reviewer for highlighting this relevant work. The study by [6] shows that an LLM agent with a BDI structure can model human trust behavior in static, controlled settings. Building on this foundation, our work embeds a similar BDI loop into a dynamic multi-agent simulation, where agents iteratively revise their beliefs in response to evolving news and peer interactions. We will cite this paper and clarify this distinction in the next version.
>
> [6] Xie, C, et al. (2024). Can large language model agents simulate human trust behavior?. Advances in neural information processing systems, 37, 15674-15729.
>
> > Weakness 2 & Question 3: quantitative benchmarking
>
> We thank the reviewer for this insightful suggestion regarding the need for more rigorous quantitative benchmarking beyond point metrics. Following your advice, we have incorporated confidence interval analysis into our evaluation, which further strengthens the empirical rigor of our results. Specifically, we follow the statistical methodology of [7] and apply block bootstrap tests to construct 95% confidence intervals (CI) for the three quantitatively measurable **Stylized Facts (SF)** in Sec. 4.2: SF I (Kurtosis), SF II (Leverage Effect), and SF IV (Volatility Clustering). The resulting CI, based on real data, are summarized below:
>
> |Stylized Facts|SF I (Kurtosis)|SF II (Leverage Effect)|SF IV (Volatility Clustering)|
> |-|-|-|-|
> |95% CI|[5.23, 8.74]|[0.003, 0.16]|[0.2529, 0.9018]|
>
> According to the numerical results from seven simulation runs using different backbone models (summarized in Appendix D.3, Table 9), we find that the results for SF II and SF IV consistently fall within the empirical 95% confidence intervals, indicating strong alignment with real market behavior. For SF I, although all simulated kurtosis values are greater than 3, confirming the presence of fat tails, only one run (using GPT-4o) falls within the empirical interval. This suggests that while our model captures the qualitative nature of fat-tailedness, it slightly underestimates the severity of real-world tail risks. This limitation highlights a valuable direction for future research, such as incorporating greater agent heterogeneity. Overall, the distribution of results across the seven runs shows close agreement with empirical benchmarks, further supporting the realism and robustness of our framework.
>
> [7] Franke, R., et al. (2012). Structural stochastic volatility in asset pricing dynamics: Estimation and model contest. Journal of Economic Dynamics and Control, 36(8), 1193-1211.
>
> > Typos
>
> Thank you for catching the typo. We will correct the sentence in the next version.
>
> **Best regards,**
>
> **Authors**

---

> ### Author Response · Authors · 2025-08-07
> **Looking forward to further discussion**
>
> **Dear Reviewer 4qQH,**
>
> We are writing to express our sincere gratitude for your thorough and insightful review of our manuscript. We genuinely appreciate your positive feedback and were particularly encouraged by your comments highlighting the ambitious scope of our framework in incorporating social interactions and the thoroughness of our extensive experiments. Your validation of our work was incredibly helpful and motivating.
>
> As the discussion deadline is quickly approaching, we wanted to gently follow up to see if you have any remaining questions or if our responses have adequately addressed your comments. We are eager to ensure all your points have been fully considered.
>
> Thank you once again for your time and valuable contributions to this process.
>
> **Best regards,**
>
> **Authors**

---

> ### Comment · Area_Chair_Wze4 · 2025-08-09
>
> Dear Reviewer 4qQH
>
> Could you please take a look at the authors' rebuttal and respond accordingly?
>
> Thanks, AC

---

### Note · Authors · 2025-08-12

**Dear AC, Reviewers,**

We express our sincere gratitude to you and all the reviewers for the thorough and constructive feedback. We are encouraged that our work’s strengths were recognized, including:
- **Novelty and Ambition:** The work was commended for representing an intriguing and promising new direction for applying LLM-based agents to social simulation.
- **Extensive and Solid Experimentation:** The experiments were praised for their extensiveness and solidity, including the ablation studies and the framework's scalability.
- **Realistic and Validated Simulation:** The framework was acknowledged for its ability to realistically simulate a wide range of real-world economic phenomena, with findings that align well with established theories.

A central theme throughout the review process was the call for greater empirical rigor and enhanced clarity on our methodology and ethical considerations. We have taken these points seriously and believe we have addressed the most critical concerns:

- **Ethical and Data Concerns:** We addressed questions regarding data usage by detailing our privacy-by-design philosophy. We clarified that our process uses public data to seed entirely synthetic agents that do not correspond to any real individuals and elaborated on the fair use doctrine for our academic approach.
- **Empirical Rigor and Validation:** Following insightful suggestions, we strengthened our validation by incorporating confidence interval analysis, conducting new sensitivity analyses and targeted ablation studies, and providing new analyses linking agent traits to performance.
- **Clarity and Contribution:** We provided formal definitions for our BDI framework and further articulated how our work serves as a valuable tool for the ML community, particularly as a generator of high-quality synthetic data.

We are committed to polishing the paper. The revised manuscript will integrate all additional experiments from the rebuttal, include a dedicated section on data and ethics, and incorporate all suggested revisions to improve clarity and context.

Finally, we reiterate our appreciation for the supportive feedback. We believe our work provides a valuable contribution that helps bridge computational social science and machine learning, and we are confident it will stimulate important further discussion in the community.

**Best regards,**

**Authors**

---

### Decision · Program_Chairs · 2025-09-17

**Decision:**

Accept (poster)

**Comment:**

This paper builds an LLM Agent-based Model for financial market simulation. All reviewers agree the present work studies an interesting, promising and ambitious. The experiments are also quite extensive. I recommend acceptance.